# Manifold Denoising by Nonlinear Robust Principal Component Analysis

**He Lyu, Ningyu Sha, Shuyang Qin, Ming Yan, Yuying Xie, Rongrong Wang**
Department of Computational Mathematics, Science and Engineering
Michigan State University
{lyuhe,shaningy,qinshuya,myan,xyy,wangron6}@msu.edu

## Abstract

This paper extends robust principal component analysis (RPCA) to nonlinear manifolds. Suppose that the observed data matrix is the sum of a sparse component and a component drawn from some low dimensional manifold. Is it possible to separate them by using similar ideas as RPCA? Is there any benefit in treating the manifold as a whole as opposed to treating each local region independently? We answer these two questions affirmatively by proposing and analyzing an optimization framework that separates the sparse component from the manifold under noisy data. Theoretical error bounds are provided when the tangent spaces of the manifold satisfy certain incoherence conditions. We also provide a near optimal choice of the tuning parameters for the proposed optimization formulation with the help of a new curvature estimation method. The efficacy of our method is demonstrated on both synthetic and real datasets.

## 1 Introduction

Manifold learning and graph learning are nowadays widely used in computer vision, image processing, and biological data analysis on tasks such as classification, anomaly detection, data interpolation, and denoising. In most applications, graphs are learned from the high dimensional data and used to facilitate traditional data analysis methods such as PCA, Fourier analysis, and data clustering [7, 8, 9, 15, 12]. However, the quality of the learned graph may be greatly jeopardized by outliers which cause instabilities in all the aforementioned graph assisted applications.

In recent years, several methods have been proposed to handle outliers in nonlinear data [11, 21, 3]. Despite the success of those methods, they only aim at detecting the outliers instead of correcting them. In addition, very few of them are equipped with theoretical analysis of the statistical performances. In this paper, we propose a novel non-task-driven algorithm for the mixed noise model in (1) and provide theoretical guarantees to control its estimation error. Specifically, we consider the mixed noise model as

$$\tilde{X}_i = X_i + S_i + E_i, \quad i = 1, \dots, n, \tag{1}$$

where $X_i \in \mathbb{R}^p$ is the noiseless data independently drawn from some manifold $\mathcal{M}$ with an intrinsic dimension $d \ll p$, $E_i$ is the i.i.d. Gaussian noise with small magnitudes, and $S_i$ is the sparse noise with possibly large magnitudes. If $S_i$ has a large entry, then the corresponding $\tilde{X}_i$ is usually considered as an outlier. The goal of this paper is to simultaneously recover $X_i$ and $S_i$ from $\tilde{X}_i$, $i = 1, .., n$.

There are several benefits in recovering the noise term $S_i$ along with the signal $X_i$. First, the support of $S_i$ indicates the locations of the anomaly, which is informative in many applications. For example, if $X_i$ is the gene expression data from the $i$th patient, the nonzero elements in $S_i$ indicate the differentially expressed genes that are the candidates for personalized medicine. Similarly, if $S_i$ is a

result of malfunctioned hardware, its nonzero elements indicate the locations of the malfunctioned parts. Secondly, the recovery of $S_i$ allows the "outliers" to be pulled back to the data manifold instead of simply being discarded. This prevents a waste of information and is especially beneficial in cases where data is insufficient. Thirdly, in some applications, the sparse $S_i$ is a part of the clean data rather than a noise term, then the algorithm provides a natural decomposition of the data into a sparse and a non-sparse component that may carry different pieces of information.

Along a similar line of research, Robust Principle Component Analysis (RPCA) [2] has received considerable attention and has demonstrated its success in separating data from sparse noise in many applications. However, its assumption that the data lies in a low dimensional subspace is somewhat strict. In this paper, we generalize the Robust PCA idea to the non-linear manifold setting. The major new components in our algorithm are: 1) an incorporation of the manifold curvature information into the optimization framework, and 2) a unified way to apply RPCA to a collection of tangent spaces of the manifold.

## 2 Methodology

Let $\tilde{X} = [\tilde{X}_1, \ldots, \tilde{X}_n] \in \mathbb{R}^{p \times n}$ be the noisy data matrix containing $n$ samples. Each sample is a vector in $\mathbb{R}^p$ independently drawn from (1). The overall data matrix $\tilde{X}$ has the representation

$$\tilde{X} = X + S + E$$

where $X$ is the clean data matrix, $S$ is the matrix of the sparse noise, and $E$ is the matrix of the Gaussian noise. We further assume that the clean data $X$ lies on some manifold $\mathcal{M}$ embedded in $\mathbb{R}^p$ with a small intrinsic dimension $d \ll p$ and the samples are sufficient ($n \geq p$). The small intrinsic dimension assumption ensures that data is locally low dimensional so that the corresponding local data matrix is of low rank. This property allows the data to be separated from the sparse noise.

The key idea behind our method is to handle the data locally. We use the $k$ Nearest Neighbors ($k$NN) to construct local data matrices, where $k$ is larger than the intrinsic dimension $d$. For a data point $X_i \in \mathbb{R}^p$, we define the local patch centered at it to be the set consisted of its $k$NN and itself, and a local data matrix $X^{(i)}$ associated with this patch is $X^{(i)} = [X_{i_1}, X_{i_2}, \ldots, X_{i_k}, X_i]$, where $X_{i_j}$ is the $j$th-nearest neighbor of $X_i$. Let $\mathcal{P}_i$ be the restriction operator to the $i$th patch, i.e., $\mathcal{P}_i(X) = XP_i$ where $P_i$ is the $n \times (k+1)$ matrix that selects the columns of $X$ in the $i$th patch. Then $X^{(i)} = \mathcal{P}_i(X)$. Similarly, we define $S^{(i)} = \mathcal{P}_i(S)$, $E^{(i)} = \mathcal{P}_i(E)$ and $\tilde{X}^{(i)} = \mathcal{P}_i(\tilde{X})$.

Since each local data matrix $X^{(i)}$ is nearly of low rank and $S$ is sparse, we can decompose the noisy data matrix into low-rank parts and sparse parts through solving the following optimization problem

$$\{\hat{S}, \{\hat{S}^{(i)}\}_{i=1}^n, \{\hat{L}^{(i)}\}_{i=1}^n\} = \underset{S, S^{(i)}, L^{(i)}}{\arg\min} F(S, \{S^{(i)}\}_{i=1}^n, \{L^{(i)}\}_{i=1}^n)$$

$$\equiv \underset{S, S^{(i)}, L^{(i)}}{\arg\min} \sum_{i=1}^n \left( \lambda_i \|\tilde{X}^{(i)} - L^{(i)} - S^{(i)}\|_F^2 + \|\mathcal{C}(L^{(i)})\|_* + \beta \|S^{(i)}\|_1 \right)$$

$$\text{subject to } S^{(i)} = \mathcal{P}_i(S), \tag{2}$$

here we take $\beta = \max\{k+1, p\}^{-1/2}$ as in RPCA, $\tilde{X}^{(i)} = \mathcal{P}_i(\tilde{X})$ is the local data matrix on the $i$th patch and $\mathcal{C}$ is the centering operator that subtracts the column mean: $\mathcal{C}(Z) = Z(I - \frac{1}{k+1}\mathbf{1}\mathbf{1}^T)$, where $\mathbf{1}$ is the $(k+1)$-dimensional column vector of all ones. Here we are decomposing the data on each patch into a low-rank part $L^{(i)}$ and a sparse part $S^{(i)}$ by imposing the nuclear norm and entry-wise $\ell_1$ norm on $L^{(i)}$ and $S^{(i)}$, respectively. There are two key components in this formulation: 1). the local patches are overlapping (for example, the first data point $X_1$ may belong to several patches). Thus, the constraint $S^{(i)} = \mathcal{P}_i(S)$ is particularly important because it ensures copies of the same point on different patches (and those of the sparse noise on different patches) remain the same. 2). we do not require $L^{(i)}$ to be restrictions of a universal $L$ to the $i$th patch, because the $L^{(i)}$s correspond to the local affine tangent spaces, and there is no reason for a point on the manifold to have the same projection on different tangent spaces. This seemingly subtle difference has a large impact on the final result.

If the data only contains sparse noise, i.e., $E = 0$, then $\hat{X} \equiv \tilde{X} - \hat{S}$ is the final estimation for $X$. If $E \neq 0$, we apply Singular Value Hard Thresholding [6] to truncate $\mathcal{C}(\tilde{X}^{(i)} - \mathcal{P}_i(S))$ and remove

the Gaussian noise (See §6), and use the resulting $\hat{L}_{\tau^*}^{(i)}$ to construct a final estimate $\hat{X}$ of $X$ via least squares fitting

$$\hat{X} = \arg\min_{Z \in \mathbb{R}^{p \times n}} \sum_{i=1}^{n} \lambda_i \|\mathcal{P}_i(Z) - \hat{L}_{\tau^*}^{(i)}\|_F^2. \tag{3}$$

The following discussion revolves around (2) and (3), and the structure of the paper is as follows. In §3, we explain the geometric meaning of each term in (2). In §4, we establish theoretical recovery guarantees for (2) which justifies our choice of $\beta$ and allows us to theoretically choose $\lambda$. The calculation of $\lambda$ uses the curvature of the manifold, so in §5, we provide a simple method to estimate the average manifold curvature and the method is robust to sparse noise. The optimization algorithms that solve (2) and (3) are presented in §6 and the numerical experiments are in §7.

## 3 Geometric explanation

We provide a geometric intuition for the formulation (2). Let us write the clean data matrix $X^{(i)}$ on the $i$th patch in its Taylor expansion along the manifold,

$$X^{(i)} = X_i 1^T + T^{(i)} + R^{(i)}, \tag{4}$$

where the Taylor series is expanded at $X_i$ (the center point of the $i$th patch), $T^{(i)}$ stores the first order term and its columns lie in the tangent space of the manifold at $X_i$, and $R^{(i)}$ contains all the higher order terms. The sum of the first two terms $X_i 1^T + T^{(i)}$ is the linear approximation to $X^{(i)}$ that is unknown if the tangent space is not given. This linear approximation precisely corresponds to the $L^{(i)}$s in (2), i.e., $L^{(i)} = X_i 1^T + T^{(i)}$. Since the tangent space has the same dimensionality $d$ as the manifold, with randomly chosen points, we have with probability one, that $\text{rank}(T^{(i)}) = d$. As a result, $\text{rank}(L^{(i)}) = \text{rank}(X_i 1^T + T^{(i)}) \le d + 1$. By the assumption that $d < \min\{p, k\}$, we know that $L^{(i)}$ is indeed low rank.

Combing (4) with $\tilde{X}^{(i)} = X^{(i)} + S^{(i)} + E^{(i)}$, we find the misfit term $\tilde{X}^{(i)} - L^{(i)} - S^{(i)}$ in (2) equals $E^{(i)} + R^{(i)}$. This implies that the misfit contains the high order residues (i.e., the linear approximation error) and the Gaussian noise.

## 4 Theoretical choice of tuning parameters

To establish the error bound, we need a coherence condition on the tangent spaces of the manifold.

**Definition 4.1** *Let $U \in \mathbb{R}^{m \times r}$ $(m \ge r)$ be a matrix with $U^*U = I$, the coherence of $U$ is defined as*

$$\mu(U) = \frac{m}{r} \max_{k \in \{1, \dots, m\}} \|U^* \mathbf{e}_k\|_2^2,$$

*where $\mathbf{e}_k$ is the $k$th element of the canonical basis. For a subspace $T$, its coherence is defined as*

$$\mu(V) = \frac{m}{r} \max_{k \in \{1, \dots, m\}} \|V^* \mathbf{e}_k\|_2^2,$$

*where $V$ is an orthonormal basis of $T$. The coherence is independent of the choice of basis.*

The following theorem is proved for local patches constructed using the $\epsilon$-neighborhoods. We use $k$NN in the experiments because $k$NN is more robust to insufficient samples. The full version of Theorem 4.2 can be found in the supplementary material.

**Theorem 4.2** *[succinct version] Let each $X_i \in \mathbb{R}^p$, $i = 1, \dots, n$, be independently drawn from a compact manifold $\mathcal{M} \subseteq \mathbb{R}^p$ with an intrinsic dimension $d$ and endowed with the uniform distribution. Let $X_{i_j}$, $j = 1, \dots, k_i$ be the $k_i$ points falling in an $\eta$-neighborhood of $X_i$ with radius $\eta$, where $\eta > 0$ is some fixed small constant. These points form the matrix $X^{(i)} = [X_{i_1}, \dots, X_{i_{k_i}}, X_i]$. For any $q \in \mathcal{M}$, let $T_q$ be the tangent space of $\mathcal{M}$ at $q$ and define $\bar{\mu} = \sup_{q \in \mathcal{M}} \mu(T_q)$. Suppose the support of the noise matrix $S^{(i)}$ is uniformly distributed among all sets of cardinality $m_i$. Then as long as $d < \rho_r \min\{\underline{k}, p\} \bar{\mu}^{-1} \log^{-2} \max\{\bar{k}, p\}$, and $m_i \le 0.4\rho_s p\underline{k}$ (here $\rho_r$ and $\rho_s$ are positive constants,*

$\bar{k} = \max_i k_i$, and $\underline{k} = \min_i k_i$) , *then with probability over* $1 - c_1 n \max\{\underline{k}, p\}^{-10} - e^{-c_2 \underline{k}}$ *for some constants $c_1$ and $c_2$, the minimizer $\hat{S}$ to (2) with weights*

$$\lambda_i = \frac{\min\{k_i + 1, p\}^{1/2}}{\epsilon_i}, \quad \beta_i = \max\{k_i + 1, p\}^{-1/2} \tag{5}$$

*has the error bound*

$$\sum_i \|\mathcal{P}_i(\hat{S}) - S^{(i)}\|_{2,1} \le C\sqrt{pn\bar{k}}\|\epsilon\|_2.$$

*Here $\epsilon_i = \|\tilde{X}^{(i)} - X_i 1^T - T^{(i)} - S^{(i)}\|_F$ will be estimated in the next section, $\epsilon = [\epsilon_1, ..., \epsilon_n]$, $\|\cdot\|_{2,1}$ stands for taking $\ell_2$ norm along columns and $\ell_1$ norm along rows, and $T^{(i)}$ is the projection of $X^{(i)} - X_i 1^T$ to the tangent space $T_{X_i}$.*

**Remark.** We can interpret $\epsilon$ as the total noise in the data. As explained in §3, $\|\tilde{X}^{(i)} - X_i 1^T - T^{(i)} - S^{(i)}\|_F = \|R^{(i)} + E^{(i)}\|_F$, thus $\epsilon = 0$ if the manifold is linear and the Gaussian noise is absent. The factor $\sqrt{n}$ in front of $\|\epsilon\|_2$ takes into account the use of different norms on the two hand sides (the right hand side is the Frobenius norm of the noise matrix $\{R^{(i)} + E^{(i)}\}_{i=1}^n$ obtained by stacking the $R^{(i)} + E^{(i)}$ associated with each patch into one big matrix). The factor $\sqrt{p}$ is due to the small weight $\beta_i$ of $\|S^{(i)}\|_1$ compared to the weight 1 on $\|\tilde{X}^{(i)} - L^{(i)} - S^{(i)}\|_F^2$. The factor $\bar{k}$ appears because on average, each column of $\hat{S} - S$ is added about $k := \frac{1}{n}\sum_i k_i$ times on the left hand side.

## 5 Estimating the curvature

The definition $\lambda_i$ in (5) involves an unknown quantity $\epsilon_i^2 = \|\tilde{X}^{(i)} - X_i 1^T - T^{(i)} - S^{(i)}\|_F^2 \equiv \|R^{(i)} + E^{(i)}\|_F^2$. We assume the standard deviation $\sigma$ of the i.i.d. Gaussian entries of $E^{(i)}$ is known, so $\|E^{(i)}\|_F^2$ can be approximated. Since $R^{(i)}$ is independent of $E^{(i)}$, the cross term $\langle R^{(i)}, E^{(i)} \rangle$ is small. Our main task is estimating $\|R^{(i)}\|_F^2$, the linear approximation error defined in §3. At local regions, second order terms dominates the linear approximation residue, hence estimating $\|R^{(i)}\|_F^2$ requires the curvature information.

### 5.1 A short review of related concepts in Riemannian geometry

The principal curvatures at a point on a high dimensional manifold are defined as the singular values of the second fundamental forms [10]. As estimating all the singular values from the noisy data may not be stable, we are only interested in estimating the mean curvature, that is the root mean squares of the principal curvatures.

For the simplicity of illustration, we review the related concepts using the 2D surface $\mathcal{M}$ embedded in $\mathbb{R}^3$ (Figure 1). For any curve $\gamma(s)$ in $\mathcal{M}$ parametrized by arclength with unit tangent vector $t_\gamma(s)$, its curvature is the norm of the covariant derivative of $t_\gamma$: $\|dt_\gamma(s)/ds\| = \|\gamma''(s)\|$. In particular, we have the following decomposition

$$\gamma''(s) = k_g(s)\hat{v}(s) + k_n(s)\hat{n}(s),$$

where $\hat{n}(s)$ is the unit normal direction of the manifold at $\gamma(s)$ and $\hat{v}$ is the direction perpendicular to $\hat{n}(s)$ and $t_\gamma(s)$, i.e., $\hat{v} = \hat{n} \times t_\gamma(s)$. The coefficient $k_n(s)$ along the normal direction is called the normal curvature, and the coefficient $k_g(s)$ along the perpendicular direction $\hat{v}$ is called the geodesic curvature. The principal curvatures purely depend on $k_n$. In particular, in 2D, the principal curvatures are precisely the maximum and minimum of $k_n$ among all possible directions.

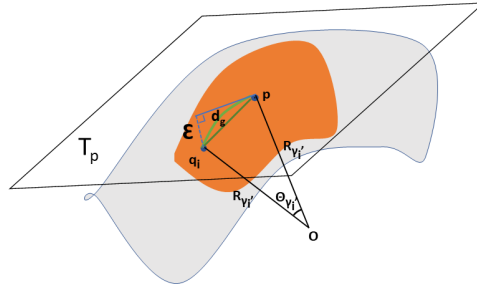

Figure 1: Local manifold geometry

A natural way to compute the normal curvature is through geodesic curves. The geodesic curve between two points is the shortest curve connecting them. Therefore geodesic curves are usually viewed as "straight lines" on the manifold. The geodesic curves have the favorable property that their curvatures have 0 contribution from $k_g$. That is to say, the second order derivative of the geodesic curve parameterized by the arclength is exactly $k_n$.

---

**Algorithm 1:** Estimate the mean curvature $\bar{\Gamma}(p)$ at some point $p$

---

**Input:** Distance matrix $D$, adjacency matrix $A$, some proper constants $r_1 < r_2$, number of pairs $m$
**Output:** the estimated mean curvature $\bar{\Gamma}(p)$

1 **for** *i = 1: m* **do**
2      Randomly pick some point $q_i \in B(p, r_2) \backslash B(p, r_1)$;
3      Calculate the geodesic distance $d_g(p, q_i)$ using $A$;
4      Solve for the radius $R_i$ based on (7);
5 **end**
6 Compute estimated curvature $\bar{\Gamma}(p) = (\frac{1}{m} \sum_{i=1}^{m} R_i^{-2})^{1/2}$.

---

---

**Algorithm 2:** Estimate the overall curvature $\bar{\Gamma}(\Omega)$ for some region $\Omega$

---

**Input:** Distance matrix $D$, adjacency matrix $A$, some proper constants $r_1 < r_2$, number of pairs $m$
**Output:** the estimated overall curvature $\bar{\Gamma}(\Omega)$

1 **for** *i = 1: m* **do**
2      Randomly pick a pair of points $p_i, q_i \in \Omega$ such that $r_1 \leq d(p_i, q_i) \leq r_2$ ;
3      Calculate the geodesic distance $d_g(p_i, q_i)$ using $A$;
4      Solve for the radius $R_i$ based on (7);
5 **end**
6 Compute estimated curvature $\bar{\Gamma}(\Omega) = (\frac{1}{m} \sum_{i=1}^{m} R_i^{-2})^{1/2}$.

---

## 5.2 The proposed method

All existing curvature estimation methods we are aware of are in the field of computer vision where the objects are 2D surfaces in 3D [5, 4, 19, 14]. Most of these methods are difficult to generalize to high ($> 3$) dimensions with the exception of the integral invariant based approaches [17]. However, the integral invariant based approaches is not robust to sparse noise and is unsuited to our problem.

We propose a new method to estimate the mean curvature from the noisy data. Although the graphic illustration is made in 3D, the method is dimension independent. To compute the average normal curvature at a point $p \in \mathcal{M}$, we randomly pick $m$ points $q_i \in \mathcal{M}$ on the manifold lying within a proper distance to $p$ as specified in Algorithm 1. Let $\gamma_i$ be the geodesic curve between $p$ and $q_i$. For each $i$, we compute the pairwise Euclidean distance $\|p - q_i\|_2$ and the pairwise geodesic distance $d_g(p, q_i)$ using the Dijkstra's algorithm. Through a circular approximation of the geodesic curve as drawn in Figure 1, we can compute the curvature of the geodesic curve as the inverse of the radius

$$\|\gamma_i''(p)\| = 1/R_{\gamma_i'}, \tag{6}$$

where $\gamma_i'$ is the tangent direction along which the curvature is calculated and $R_{\gamma_i'}$ is the radius of the circular approximation to the curve $\gamma$ at $p$, which can be solved along with the angle $\theta_{\gamma_i'}$ through the geometric relations

$$2R_{\gamma_i'} \sin(\theta_{\gamma_i'}/2) = \|p - q_i\|_2, \quad R_{\gamma_i'} \theta_{\gamma_i'} = d_g(p, q_i), \tag{7}$$

as indicated in Figure 1. Finally, we define the average curvature $\bar{\Gamma}(p)$ at $p$ to be

$$\bar{\Gamma}(p) := (\mathbb{E}_{q_i} \|\gamma_i''(p)\|^2)^{1/2} \equiv (\mathbb{E}_{q_i} R_{\gamma_i}^{-2})^{1/2}. \tag{8}$$

To estimate the mean curvature from the data, we construct two matrices $D$ and $A$. $D \in \mathbb{R}^{n \times n}$ is the pairwise distance matrix, where $D_{ij}$ denotes the Euclidean distance between two points $X_i$ and $X_j$. $A$ is a type of adjacency matrix defined as follows and is to be used to compute the pairwise geodesic distances from the data,

$$A_{ij} = \begin{cases} D_{ij} & \text{if } X_j \text{ is in the } k \text{ nearest neighbors of } X_i \\ 0 & \text{elsewhere.} \end{cases} \tag{9}$$

Algorithm 1 estimates the mean curvature at some point $p$ and Algorithm 2 estimates the overall curvature within some region $\Omega$ on the manifold.

The geodesic distance is computed using the Dijkstra's algorithm, which is not accurate when $p$ and $q$ are too close to each other. The constant $r_1$ in Algorithm 1 and 2 is thus used to make sure that $p$ and $q$ are sufficiently apart. The constant $r_2$ is to make sure that $q$ is not too far away from $p$, as after all we are computing the mean curvature around $p$.

### 5.3 Estimating $\lambda_i$ from the mean curvature

We provide a way to approximate $\lambda_i$ when the number of points $n$ is finite. In the asymptotic limit $(k \to \infty, k/n \to 0)$, all the approximate sign "$\approx$" below become "$=$".

Fix a point $p \in \mathcal{M}$ and another point $q_i$ in the $\eta$-neighborhood of $p$. Let $\gamma_i$ be the geodesic curve between them. With the computed curvature $\bar{\Gamma}(p)$, we can estimate the linear approximation error of expanding $q_i$ at $p$: $q_i \approx p + P_{T_p}(q_i - p)$, where $P_{T_p}$ is the projection onto the tangent space at $p$. Let $\mathcal{E}$ be the error of this linear approximation $\mathcal{E}(q_i, p) = q_i - p - P_{T_p}(q_i - p) = P_{T_p^\perp}(q_i - p)$ where $T_p^\perp$ is the orthogonal complement of the tangent space. From Figure 1, the relation between $\|\mathcal{E}(p, q_i)\|_2$, $\|p - q_i\|_2$, and $\theta_{\gamma_i'}$ is

$$\|\mathcal{E}(p, q_i)\|_2 \approx \|p - q_i\|_2 \sin \frac{\theta_{\gamma_i'}}{2} = \frac{\|p - q_i\|_2^2}{2 R_{\gamma_i'}}. \tag{10}$$

To obtain a closed-form formula for $\mathcal{E}$, we assume that for the fixed $p$ and a randomly chosen $q_i$ in an $\xi$ neighborhood of $p$, the projection $P_{T_p}(q_i - p)$ follows a uniform distribution in a ball with radius $\eta'$ (in fact $\eta' \approx \eta$ since when $\eta$ is small, the projection of $q - p$ is almost $q - p$ itself, therefore the radius of the projected ball almost equal to the radius of the original neighborhood). Under this assumption, let $r_i = \|P_{T_p}(q_i - p)\|_2$ be the magnitude of the projection and $\phi_i = P_{T_p}(q_i - p)/\|P_{T_p}(q_i - p)\|_2$ be the direction, by [20], $r_i$ and $\phi_i$ are independent of each other. As the curvature $R_{\gamma_i}$ only depends on the direction, the numerator and the denominator of the right hand side of (10) are independent of each other. Therefore,

$$\mathbb{E}\|\mathcal{E}(p, q_i)\|_2^2 \approx \mathbb{E}\frac{\|p - q_i\|_2^4}{4 R_{\gamma_i'}^2} = \frac{\mathbb{E}\|p - q_i\|_2^4}{4} \mathbb{E} R_{\gamma_i'}^{-2} = \frac{\mathbb{E}\|p - q_i\|_2^4}{4} \cdot \bar{\Gamma}^2(p), \tag{11}$$

where the first equality used the independence and the last equality used the definition of the mean curvature in the previous subsection.

Now we apply this estimation to the neighborhood of $X_i$. Let $p = X_i$, and $q_j = X_{i_j}$ be the neighbors of $X_i$. Using (11), the average linear approximation error on this patch is

$$\frac{1}{k}\|R^{(i)}\|_F^2 := \frac{1}{k} \sum_{j=1}^{k} \|\mathcal{E}(X_{i_j}, X_i)\|_2^2 \xrightarrow{k \to \infty} \frac{\mathbb{E}\|X_i - X_{i_j}\|_2^4}{4} \bar{\Gamma}^2(X_i), \tag{12}$$

where the right hand side can also be estimated with

$$\frac{1}{k} \sum_{j=1}^{k} \frac{\|X_i - X_{i_j}\|_2^4}{4} \bar{\Gamma}^2(X_i) \xrightarrow{k \to \infty} \frac{\mathbb{E}\|X_i - X_{i_j}\|_2^4}{4} \bar{\Gamma}^2(X_i) \tag{13}$$

so when $k$ is sufficient large, $\frac{1}{k}\|R^{(i)}\|_F^2$ is also close to $\frac{1}{k} \sum_{j=1}^{k} \frac{\|X_i - X_{i_j}\|_2^4}{4} \bar{\Gamma}^2(X_i)$, which can be completely computed from the data. Combining this with the argument at the beginning of §5 we get,

$$\epsilon_i = \|R^{(i)} + E^{(i)}\|_F \approx \sqrt{\|R^{(i)}\|_F^2 + \|E^{(i)}\|_F^2} \approx \left( (k+1)p\sigma^2 + \sum_{j=1}^{k} \frac{\|X_i - X_{i_j}\|_2^4}{4} \bar{\Gamma}^2(X_i) \right)^{1/2} =: \hat{\epsilon}.$$

Thus we can set $\hat{\lambda}_i = \frac{\min\{k+1, p\}^{1/2}}{\hat{\epsilon}_i}$ due to (5). We show in the supplementary material that $\left| \frac{\hat{\lambda}_i - \lambda_i^*}{\lambda_i^*} \right| \xrightarrow{k \to \infty} 0$, where $\lambda_i^* = \frac{\min\{k+1, p\}^{1/2}}{\epsilon_i}$ as in (5).

## 6 Optimization algorithm

To solve the convex optimization problem (2) in a memory-economic way, we first write $L^{(i)}$ as a function of $S$ and eliminate them from the problem. We can do so by fixing $S$ and minimizing the objective function with respect to $L^{(i)}$

$$\hat{L}^{(i)} = \arg\min_{L^{(i)}} \lambda_i \|\tilde{X}^{(i)} - L^{(i)} - S^{(i)}\|_F^2 + \|\mathcal{C}(L^{(i)})\|_*$$
$$= \arg\min_{L^{(i)}} \lambda_i \|\mathcal{C}(L^{(i)}) - \mathcal{C}(\tilde{X}^{(i)} - S^{(i)})\|_F^2 + \|\mathcal{C}(L^{(i)})\|_* + \lambda_i \|(I - \mathcal{C})(L^{(i)} - (\tilde{X}^{(i)} - S^{(i)}))\|_F^2. \tag{14}$$

Notice that $L^{(i)}$ can be decomposed as $L^{(i)} = \mathcal{C}(L^{(i)}) + (I - \mathcal{C})(L^{(i)})$, set $A = \mathcal{C}(L^{(i)}), B = (I - \mathcal{C})(L^{(i)})$, then (14) is equivalent to

$$(\hat{A}, \hat{B}) = \underset{A,B}{\arg\min} \ \lambda_i \|A - \mathcal{C}(\tilde{X}^{(i)} - S^{(i)})\|_F^2 + \|A\|_* + \lambda_i \|B - (I - \mathcal{C})(\tilde{X}^{*(i)} - S^{(i)}))\|_F^2,$$

which decouples into

$$\hat{A} = \underset{A}{\arg\min} \ \lambda_i \|A - \mathcal{C}(\tilde{X}^{(i)} - S^{(i)})\|_F^2 + \|A\|_*, \ \hat{B} = \underset{B}{\arg\min} \lambda_i \|B - (I - \mathcal{C})(\tilde{X}^{(i)} - S^{(i)})\|_F^2.$$

The problems above have closed form solutions

$$\hat{A} = \mathcal{T}_{1/2\lambda_i}(\mathcal{C}(\tilde{X}^{(i)} - \mathcal{P}_i(S))), \ \hat{B} = (I - \mathcal{C})(\tilde{X}^{(i)} - \mathcal{P}_i(S)) \tag{15}$$

where $\mathcal{T}_\mu$ is the soft-thresholding operator on the singular values

$$\mathcal{T}_\mu(Z) = U \max\{\Sigma - \mu I, 0\} V^*, \ \text{where } U\Sigma V^* \text{ is the SVD of } Z.$$

Combing $\hat{A}$ and $\hat{B}$, we have derived the closed form solution for $\hat{L}^{(i)}$

$$\hat{L}^{(i)}(S) = \mathcal{T}_{1/2\lambda_i}(\mathcal{C}(\tilde{X}^{(i)} - \mathcal{P}_i(S))) + (I - \mathcal{C})(\tilde{X}^{(i)} - \mathcal{P}_i(S)). \tag{16}$$

Plugging (16) into $F$ in (2), the resulting optimization problem solely depends on $S$. Then we apply FISTA [1, 18] to find the optimal solution $\hat{S}$ with

$$\hat{S} = \underset{S}{\arg\min} \ F(\hat{L}^{(i)}(S), S). \tag{17}$$

Once $\hat{S}$ is found, if the data has no Gaussian noise, then the final estimation for $X$ is $\hat{X} \equiv \tilde{X} - \hat{S}$; if there is Gaussian noise, we use the following denoised local patches $\hat{L}_{\tau^*}^{(i)}$

$$\hat{L}_{\tau^*}^{(i)} = H_{\tau^*}(\mathcal{C}(\tilde{X}^{(i)} - \mathcal{P}_i(\hat{S}))) + (I - \mathcal{C})(\tilde{X}^{(i)} - \mathcal{P}_i(\hat{S})), \tag{18}$$

where $H_{\tau^*}$ is the Singular Value Hard Thresholding Operator with the optimal threshold as defined in [6]. This optimal thresholding removes the Gaussian noise from $\hat{L}_{\tau^*}^{(i)}$. With the denoised $\hat{L}_{\tau^*}^{(i)}$, we solve (3) to obtain the denoised data

$$\hat{X} = (\sum_{i=1}^{n} \lambda_i \hat{L}_{\tau^*}^{(i)} P_i^T)(\sum_{i=1}^{n} \lambda_i P_i P_i^T)^{-1}. \tag{19}$$

The proposed Nonlinear Robust Principle Component Analysis (NRPCA) algorithm is summarized in Algorithm 3. There is one caveat in solving (2): the strong sparse noise may result in a wrong

---

**Algorithm 3:** Nonlinear Robust PCA

**Input:** Noisy data matrix $\tilde{X}$, $k$ (number of neighbors in each local patch), $T$ (number of neighborhood updates iterations)

**Output:** the denoised data $\hat{X}$, the estimated sparse noise $\hat{S}$

1 Estimate the curvature using (8);
2 Estimate $\lambda_i$, $i = 1, \ldots, n$ as in §5, set $\beta$ as in (2);
3 $\hat{S} \leftarrow 0$;
4 **for** *iter = 1: T* **do**
5      Find the kNN for each point using $\tilde{X} - \hat{S}$ and construct the restriction operators $\{\mathcal{P}_i\}_{i=1}^n$;
6      Construct the local data matrices $\tilde{X}^{(i)} = \mathcal{P}_i(\tilde{X})$ using $\mathcal{P}_i$ and the noisy data $\tilde{X}$;
7      $\hat{S} \leftarrow$ minimizer of (17) iteratively using FISTA;
8 **end**
9 Compute each $\hat{L}_{\tau^*}^{(i)}$ from (18) and assign $\hat{X}$ from (19).

---

neighborhood assignment when constructing the local patches. Therefore, once $\hat{S}$ is obtained and removed from the data, we update the neighborhood assignment and re-compute $\hat{S}$. This procedure is repeated $T$ times.

# 7 Numerical experiment

**Simulated Swiss roll:** We demonstrate the superior performance of NRPCA on a synthetic dataset following the mixed noise model (1). We sampled 2000 noiseless data $X_i$ uniformly from a 3D Swiss roll and generated the Gaussian noise matrix with i.i.d. entries obeying $\mathcal{N}(0, 0.25)$. The sparse noise matrix $S$ is generated by randomly replacing 100 entries of a zero $p \times n$ matrix with i.i.d. samples generated from $(-1)^y \cdot z$ where $y \sim \text{Bernoulli}(0.5)$ and $z \sim \mathcal{N}(5, 0.09)$. We applied NRPCA to the simulated data with patch size $k = 15$. Figure 2 reports the denoising results in the original space (3D) looking down from above. We compare two ways of using the outputs of NRPCA: 1). only remove the sparse noise from the data $\tilde{X} - \hat{S}$; 2). remove both the sparse and Gaussian noise from the data: $\hat{X}$. In addition, we plotted $\tilde{X} - \hat{S}$ with and without the neighbourhood update. These results are all superior to an ad-hoc application of the Robust PCA on the individual local patches.

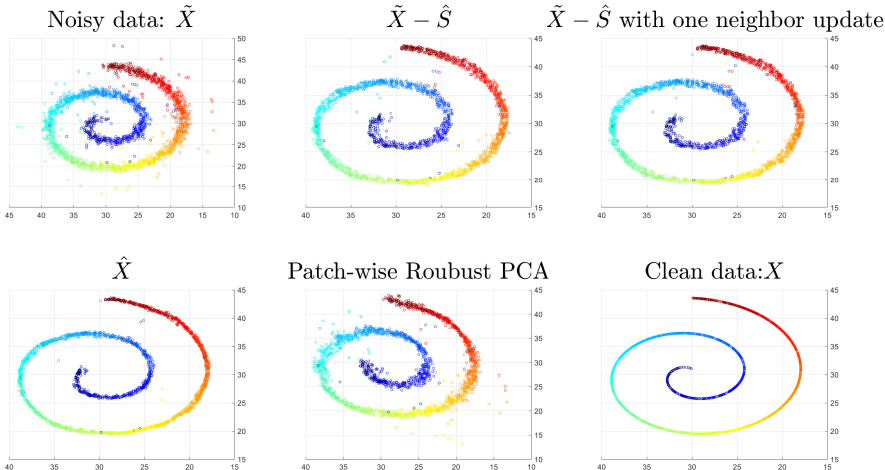

Figure 2: NRPCA applied to the noisy 3D Swiss roll dataset. $\tilde{X} - \hat{S}$ is the result after subtracting the sparse noise estimated by setting $T = 1$ in NRPCA, i.e., no neighbour update; "$\tilde{X} - \hat{S}$ with one neighbor update" used the $\hat{S}$ obtained by setting $T = 2$ in NRPCA; clearly, the neighbour update helped to remove more sparse noise; $\hat{X}$ is the data obtained via fitting the denoised tangent spaces as in (3). Compared to "$\tilde{X} - \hat{S}$ with one neighbor update", it further removed the Gaussian noise from the data; "Patch-wise Robust PCA" refers to the ad-hoc application of the vanilla Robust PCA to each local patch independently, whose performance is worse than the proposed joint-recovery formulation.

**The MNIST datasest:** We observed some interesting dimension reduction result of MNIST with the help of NRPCA. It is well-known that the handwritten digits 4 and 9 are so similar that the popular dimension reduction methods Isomap and Laplacian Eigenmaps fail to separate them into two clusters (first column of Figure 3). We conjecture that the similarity between the two clusters is caused by personalized writing styles of the beginning and finishing strokes. As this type of variation can be better modeled by sparse noise than Gaussian or Poisson noises, we applied NRPCA to the raw MNIST images. The right column of Figure 3 shows that after the NRPCA denoising (with $k = 11$), the separability of the two clusters in the first two coordinates of Isomap and Laplacian Eigenmaps increases. In addition, these new embeddings seem to suggest that some trajectory patterns exist in the data. We provide additional plots in the supplementary material to support this observation.

**Biological data:** We illustrate the potential usefulness of NRPCA algorithm on an embryoid body (EB) differentiation dataset over a 27-day time course, which consists of gene expressions for 31,000 cells measured with single-cell RNA-sequencing technology (scRNAseq) [13, 16]. This EB data comprising expression measurement for cells originated from embryoid at different stages is hence developmental in nature, which should exhibit a progressive type of characters such as tree structure because all cells arise from a single oocyte and then develop into different highly-differentiated tissues. This progression character is often missing when we directly apply dimension reduction

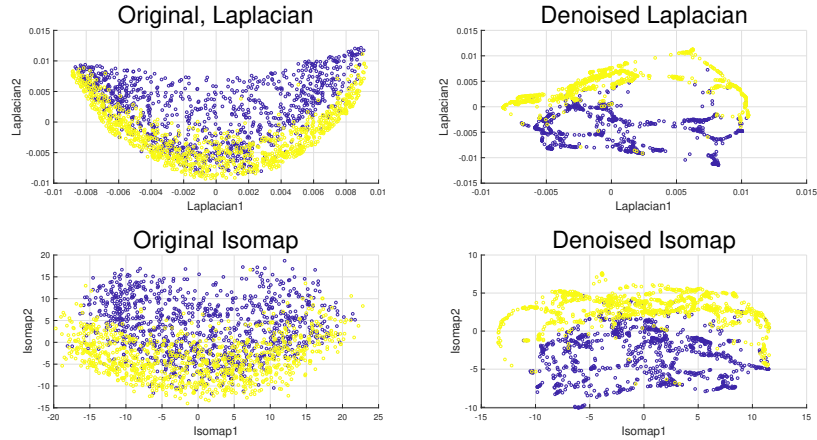

Figure 3: Laplacian eigenmaps and Isomap results for the original and the NRPCA denoised digits 4 and 9 from the MNIST dataset.

methods to the data as shown in Figure 4 because biological data including scRNAseq is highly noisy and often is contaminated with outliers from different sources including environmental effects and measurement error. In this case, we aim to reveal the progressive nature of the single-cell data from transcript abundance as measured by scRNAseq.

We first normalized the scRNAseq data following the procedure described in [16] and randomly selected 1000 cells using the stratified sampling framework to maintain the ratios among different developmental stages. We applied our NRPCA method to the normalized subset of EB data and then applied Locally Linear Embedding (LLE) to the denoised results. The two-dimensional LLE results are shown in Figure 4. Our analysis demonstrated that although LLE is unable to show the progression structure using noisy data, after the NRPCA denoising, LLE successfully extracted the trajectory structure in the data, which reflects the underlying smooth differentiating processes of embryonic cells. Interestingly, using the denoised data from $\widetilde{X} - \hat{S}$ with neighbor update, the LLE embedding showed a branching at around day 9 and increased variance in later time points, which was confirmed by manual analysis using 80 biomarkers in [16].

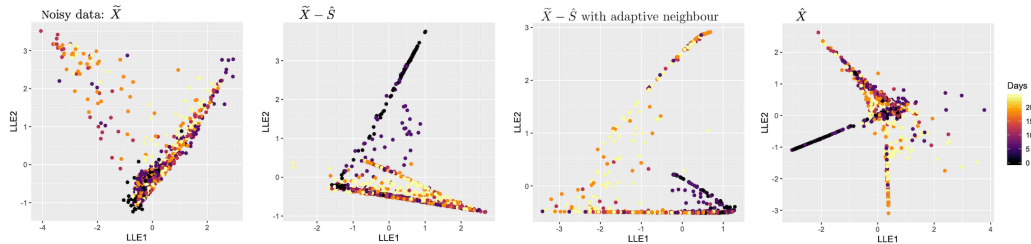

Figure 4: LLE results for denoised scRNAseq data set.

# 8   Conclusion

In this paper, we proposed the first outlier correction method for nonlinear data analysis that can correct outliers caused by the addition of large sparse noise. The method is a generalization of the Robust PCA method to the nonlinear setting. We provided procedures to treat the non-linearity by working with overlapping local patches of the data manifold and incorporating the curvature information into the denoising algorithm. We established a theoretical error bound on the denoised data that holds under conditions only depending on the intrinsic properties of the manifold. We tested our method on both synthetic and real dataset that were known to have nonlinear structures and reported promising results.

**Acknowledgements** The authors would like to thank Shuai Yuan, Hongbo Lu, Changxiong Liu, Jonathan Fleck, Yichen Lou, and Lijun Cheng for useful discussions. This work was supported in part by the NIH grants U01DE029255, 5RO3DE027399 and the NSF grants DMS-1902906, DMS-1621798, DMS-1715178, CCF-1909523 and NCS-1630982.

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
