[Supplementary Material]

# Supplementary Material for "Manifold Denoising by Nonlinear Robust Principal Component Analysis"

**He Lyu, Ningyu Sha, Shuyang Qin, Ming Yan, Yuying Xie, Rongrong Wang**
Department of Computational Mathematics, Science and Engineering
Michigan State University
{lyuhe,shaningy,qinshuya,myan,xyy,wangron6}@msu.edu

This document contains the proof of Theorem 4.2, the proof of $|\frac{\hat{\lambda}_i - \lambda_i^*}{\lambda_i^*}| \xrightarrow{k\to\infty} 0$ in §5 and some supplementary numerical simulations.

## 1 Proof of Theorem 4.2

**Definition 1.** *Let $\mathcal{M}$ be a compact manifold endowed with a continuous measure $\mu$. For any $z \in \mathcal{M}$, its $(\eta(\tau), \tau)$-neighborhood $\mathcal{N}$ is the neighbourhood with radius $\eta(\tau)$ and measure $\tau$, i.e., $\mu(\mathcal{N}) = \tau$, $\mathcal{N} = \mathcal{M} \cap B_2(z, \eta(\tau))$, and $\eta(\tau) = \min\{r : \mu(\mathcal{M} \cap B_2(z, r)) = \tau\}$.*

Since $\mathcal{M}$ is compact, its measure is finite, say $\mu(\mathcal{M}) = A$, and the radii of all the $\tau$-neighbourhoods are bounded by some constant $\eta$:

$$\sup_i |\eta_i|^2 \le \eta^2.$$

**Theorem 2** (Full version of Theorem 4.2). *Given the dataset $X = [X_1, X_2, \cdots, X_n]$, let each $X_i$ be independently drawn from a compact manifold $\mathcal{M} \subseteq \mathbb{R}^p$ with intrinsic dimension $d$ and endowed with the uniform distribution $\mu$. Fix some $q > 0$, let $X_{i_j}$, $j = 1, \ldots, k_i$ be the $k_i$ points falling in the $(\eta_i, q)$-neighbourhood of $X_i$. Together they form a matrix $X^{(i)} = [X_{i_1}, \ldots, X_{i_{k_i}}, X_i]$. Suppose the i.i.d. projections $y_{i,j} \equiv P_{T_{X_i}}(X_{i_j} - X_i)$ where $T_{X_i}$ is the tangent space at $X_i$ obey the same distribution as some $a_i$ for all $j$, i.e., $y_{i,j} \sim a_i$ ($\sim$ means the two vectors are identically distributed), and the matrix $\mathbb{E}(a_i - \mathbb{E}a_i)(a_i - \mathbb{E}a_i)^*$ has a finite condition number for each $i$. In addition, suppose the support of the noise matrix $S^{(i)}$ is uniformly distributed among all sets of cardinality $m_i$. For any $\zeta \in \mathcal{M}$, let $T_\zeta$ be the tangent space of $\mathcal{M}$ at $\zeta$ and define $\mu_1 := \sup_{\zeta \in \mathcal{M}} \mu(T_\zeta)$. Then as long as $qn \ge c \log n$, $d < \rho_r \min\{nq/2, p\}\mu_1^{-1} \log^{-2} \max\{2nq, p\}$, and $\frac{m_i}{pk_i} \le 0.4\rho_s$ (here $c$, $\rho_r$ and $\rho_s$ are positive numerical constants), then with probability over $1 - c_1(n \max\{nq/2, p\}^{-10} + \exp(-c_2 nq))$ for some constants $c_1$ and $c_2$, the minimizer $\hat{S}$ to (2) with $\lambda_i = \frac{\min\{k_i+1,p\}^{1/2}}{\epsilon_i}$, and $\beta_i = \max\{k_i + 1, p\}^{-1/2}$ has the error bound*

$$\sum_i \|\mathcal{P}_i(\hat{S}) - S^{(i)}\|_{2,1} \le C\sqrt{pn\bar{k}}\|\epsilon\|_2.$$

*Here $\bar{k} = \max_i k_i$ satisfing $nq/2 \le \bar{k} \le 2nq$, $\epsilon_i = \|\tilde{X}^{(i)} - X_i 1^T - T^{(i)} - S^{(i)}\|_F$, $\epsilon = [\epsilon_1, ..., \epsilon_n]$, $\|\cdot\|_{2,1}$ stands for taking $\ell_2$ norm along columns and $\ell_1$ norm along the rows, and $T^{(i)}$ is the projection of $X^{(i)} - X_i$ to the tangent space $T_{X_i}$.*

The proof the Theorem 2 uses similar techniques as [3]. The main difference is that in [3], both the left and the right singular vectors of the data matrix are required to satisfy the coherence conditions, while here we show that only the left singular vectors that corresponding to the tangent spaces are relevant. In other words, the recovery guarantee is built solely upon assumptions on the intrinsic properties of the manifold, i.e., the tangent spaces.

**The proof architecture is as follows**. In Section 1.1, we derive the error bound in Theorem 4.2 under a small coherence condition for both the left and the right singular vectors of $L^{(i)}$. In Section 1.2, we show that the requirement on the right singular vectors can be removed using the i.i.d. assumption on the samples.

## 1.1 Deriving the error bound in Theorem 2 under coherence conditions on both the right and the left singular vectors

In Section 3 of the main paper, we explained that $L^{(i)} = X_i 1^T + T^{(i)}$ corresponds to the linear approximation of the $i$th patch. After the centering $\mathcal{C}(L^{(i)}) = \mathcal{C}(T^{(i)})$, one gets rid of the first term and the resulting matrix has a column span coincide with $T^{(i)}$. This indicates that the columns of $\mathcal{C}(L^{(i)})$ lie in the column space of the tangent space $span(T^{(i)})$, this also indicates that the rows of $L^{(i)}$ are in $span\{1^T, T^{(i)}\}$.

One can view the knowledge that $1^T$ is in the row space of $L^{(i)}$ as a prior knowledge of the left singular vectors of $L^{(i)}$. Robust PCA with prior knowledge is studied in [3], and we will use some of the result therein. Specifically, we adapt the dual certificate approach in [3] to our problem to derive the error bound for our new problem in the theorem, and choose proper $\lambda_i$, $i = 1, 2, \cdots, n$ and $\beta_i$ accordingly.

We first state the following assumptions as from [3]:

**Assumption A**

In each local patch, $L^{(i)} \in \mathbb{R}^{p,k_i+1}$, denote $n_{(1)} = \max\{p, k_i + 1\}$, $n_{(2)} = \min\{p, k_i + 1\}$, let

$$\mathcal{C}(L^{(i)}) = U_i \Sigma_i V_i^*,$$

be the singular value decomposition for each $L^{(i)}$, where $U_i \in R^{p \times d}, \Sigma_i \in \mathbb{R}^{d \times d}, V_i^* \in \mathbb{R}^{d \times (k+1)}$. let $\tilde{V}_i$ be the orthonormal basis of $span\{1, V_i\}$, assume for each $i \in \{1, 2, \cdots, n\}$, the following hold with a constant $\rho_r$ that is small enough

$$\max_j \|U_i^* e_j\|^2 \leq \frac{\rho_r d}{p}, \tag{1}$$

$$\max_j \|\tilde{V}_i^* e_j\|^2 \leq \frac{\rho_r d}{k_i}, \tag{2}$$

$$\max_j \|U_i V_i^*\|_\infty \leq \sqrt{\frac{\rho_r d}{p k_i}}. \tag{3}$$

and $\rho_r, \rho_s, p, k_i$ satisfies the following assumptions:

**Assumption B([3], Assumption III.2.)**

(a) $\rho_r \leq \min\{10^{-4}, C_1\}$,

(b) $\rho_s \leq \min\{1 - 1.5b_1(\rho_r), 0.0156\}$,

(c) $n_{(1)} \geq \max\{C_2(\rho_r), 1024\}$,

(d) $n_{(2)} \geq 100 \log^2 n_{(1)}$,

(e) $\frac{(p+k_i)^{1/6}}{\log(p+k_i)} > \frac{10.5}{(\rho_s)^{1/6}(1-5.6561)\sqrt{\rho_s}}$,

(f) $\frac{p k_i}{500 \log n_{(1)}} > \frac{1}{\rho_s^2}$,

where $b_1(\rho_r), C_2(\rho_r)$ are some constants related to $\rho_r$.

Denote $\Pi_i$ as the linear space of matrices for each local patch (note that this is different from the tangent space $T^i$ of the manifold)

$$\Pi_i := \{U_i X^* + Y \tilde{V}_i^*, X \in \mathbb{R}^{p,d}, Y \in \mathbb{R}^{p,d+1}\}.$$

As shown by [3], the following lemma holds, indicating that if incoherence condition is satisfied, then with high probability, there exists desirable dual certificate $(W, F)$.

**Lemma 3** ([3], Lemma V.8, Lemma V.9). *For fixed $i = 1, 2, \cdots, n$, if assumptions (1), (2), (3), Assumption B and other assumptions in Theorem 2 hold, then with probability at least $1 - cn_{(1)}^{-10}$, $\|\mathcal{P}_{\Omega_i}\mathcal{P}_{\Pi_i}\| \leq 1/4$, where $\Omega_i$ is the support set of $S^{(i)}$, and $\beta < \frac{3}{10}$. In addition, there exists a pair $(W_i, F_i)$ obeying*

$$U_i V_i^* + W_i = \beta(sgn(S^{(i)}) + F_i + \mathcal{P}_{\Omega_i} D_i), \tag{4}$$

*with*

$$\mathcal{P}_{\Pi_i} W_i = 0, \ \|W_i\| \leq \frac{9}{10}, \ \mathcal{P}_{\Omega_i} F_i = 0, \ \|F_i\|_\infty \leq \frac{9}{10}, \ \|\mathcal{P}_{\Omega_i} D_i\|_F \leq \frac{1}{4}. \tag{5}$$

Therefore, by union bound, with probability over $1 - cnn_{(1)}^{-10}$, for each local patch, there exists a pair $(W_i, F_i)$ obeying (4) and (5).

In Section 9.1.2, we will show that with our assumption that data is independently drawn from a manifold $\mathcal{M} \subseteq \mathbb{R}^p$ with intrinsic dimension $d$ endowed with the uniform distribution, (2) and (3) are satisfied with high probability, so we only need Assumption B and (1), which is only related to the property of tangent space of the manifold itself.

In Lemma 5, we prove that in our setting that each $X_i$ is drawn from a manifold $\mathcal{M} \subseteq \mathbb{R}^p$ independently and uniformly, with high probability, for all $i = 1, 2, \cdots n$, $k_i$ is some integer within the range $[qn/2, 2qn]$. Now we use that to prove Theorem 2, the result is stated in the following lemma.

**Lemma 4.** *If for all local patch $i = 1, 2, \cdots, n$, there exists a pair $(W_i, F_i)$ obeying (4) and (5), then the minimizer $\hat{S}$ to (2) with $\lambda_i = \frac{\min\{k_i+1,p\}^{1/2}}{\epsilon_i}$, and $\beta_i = \max\{k_i + 1, p\}^{-1/2}$ has the error bound*

$$\sum_i \|\mathcal{P}_i(\hat{S}) - S^{(i)}\|_{2,1} \leq C\sqrt{pn\bar{k}}\|\epsilon\|_2.$$

*Here $\epsilon_i = \|\tilde{X}^{(i)} - X_i 1^T - T^{(i)} - S^{(i)}\|_F$, $\epsilon = [\epsilon_1, ..., \epsilon_n]$ is defined same as Theorem 2.*

*Proof.* To simplify notation, let's start with the problem for only one local patch:

$$\min \ \lambda\|\tilde{X} - L - S\|_F^2 + \|LG\|_* + \beta\|S\|_1. \tag{6}$$

Here $\tilde{X} \in \mathbb{R}^{p \times (k+1)}$, where $k$ denotes the number of neighbors in each local patch, $G = I - \frac{1}{k+1}11^T$ is the centering matrix, recall that the noisy data $\tilde{X}$ is $\tilde{X} = X + S + E = L + R + S + E$, $\|R + E\|_F = \|\tilde{X} - L - S\|_F \leq \epsilon$ (to be more accurate, $\epsilon_i$ for patch $i$), $X$ is the clean data on the manifold, $L$ is first order Talor approximation of $X$, $R$ is higher order terms, and $E$ denotes random noise. Also denote the solution to problem (6) as $\hat{L} = L + H_1, \hat{S} = S + H_2$. We choose $\beta = \frac{1}{\sqrt{n_{(1)}}} = \frac{1}{\sqrt{\max\{k+1,p\}}}$.

Since $\hat{L}, \hat{S}$ are the solution to (6), the following holds:

$$\lambda\|\tilde{X} - L - S\|_F^2 + \|LG\|_* + \beta\|S\|_1$$
$$\geq \lambda\|\tilde{X} - (L + H_1) - (S + H_2)\|_F^2 + \|(L + H_1)G\|_* + \beta\|S + H_2\|_1$$
$$\geq \lambda\|H_1 + H_2 - (R + E)\|_F^2 + \|LG\|_* + \langle H_1 G, UV^* + W_0\rangle + \beta\|S\|_1 + \beta\langle H_2, sgn(S) + F_0\rangle$$
$$= \lambda\|H_1 + H_2\|_F^2 + \lambda\|R + E\|_F^2 - 2\lambda\langle R + E, H_1 + H_2\rangle + \|LG\|_* + \langle H_1 G, UV^*\rangle + \beta\|S\|_1$$
$$\quad + \beta\langle H_2, sgn(S)\rangle + \|\mathcal{P}_{\Pi^\perp}(H_1 G)\|_* + \beta\|\mathcal{P}_{\Omega^\perp} H_2\|_1.$$

Here we choose $W_0$ and $F_0$ such that $\langle H_1 G, W_0\rangle = \|\mathcal{P}_{\Pi^\perp}(H_1 G)\|_*$, $\langle H_2, F_0\rangle = \|\mathcal{P}_{\Omega^\perp} H_2\|_1$ same as [1]. Note that $LG = U\Sigma V^*$, $G = I - \frac{1}{k+1}11^T$ is orthogonal projector, $LG1 = 0$ implies $V^*1 = 0$, we have

$$\langle H_1 G, UV^*\rangle = \langle H_1, UV^* G\rangle = \langle H_1, UV^*(I - \frac{1}{k+1}11^T)\rangle = \langle H_1, UV^*\rangle,$$

$$\mathcal{P}_{\Pi^\perp}(H_1 G) = (I - UU^*)H_1 G(I - \tilde{V}\tilde{V}^*) = (I - UU^*)H_1(I - \frac{1}{k+1}11^T)(I - \tilde{V}\tilde{V}^*) = \mathcal{P}_{\Pi^\perp} H_1.$$

For the second equality we use the fact that $1$ lies on the subspace spanned by $\tilde{V}$, so $(I - \tilde{V}\tilde{V}^*)1 = 0$. And for any matrix $M$, $\mathcal{P}_{\Pi^\perp} M = (I - UU^*)M(I - \tilde{V}\tilde{V}^*)$.

Denote $H = H_1 + H_2$, plug in the equations above, we obtain

$$2\lambda\langle R + E, H\rangle \geq \lambda\|H\|_F^2 + \langle H_1 + H_2, UV^*\rangle + \langle H_2, \beta sgn(S) - UV^*\rangle + \|\mathcal{P}_{\Pi^\perp}H_1\|_* + \beta\|\mathcal{P}_{\Omega^\perp}H_2\|_1$$

$$\geq \lambda\|H\|_F^2 - \|H\|_F\|UV^*\|_F + \langle H_2, W - \beta F - \beta\mathcal{P}_\Omega D\rangle + \|\mathcal{P}_{\Pi^\perp}H_1\|_* + \beta\|\mathcal{P}_{\Omega^\perp}H_2\|_1$$

$$\geq \lambda\|H\|_F^2 - \sqrt{n_{(2)}}\|H\|_F - \frac{9}{10}\|\mathcal{P}_{\Pi^\perp}H_2\|_* - \frac{9}{10}\beta\|\mathcal{P}_{\Omega^\perp}H_2\|_1 - \frac{\beta}{4}\|\mathcal{P}_\Omega H_2\|_F +$$

$$\|\mathcal{P}_{\Pi^\perp}H_1\|_* + \beta\|\mathcal{P}_{\Omega^\perp}H_2\|_1.$$

In the 3rd inequality we used

$$|\langle H_2, W\rangle| = |\langle H_2, \mathcal{P}_{\Pi^\perp}W\rangle| = |\langle\mathcal{P}_{\Pi^\perp}H_2, W\rangle| \leq \|\mathcal{P}_{\Pi^\perp}H_2\|_*\|W\| \leq \frac{9}{10}\|\mathcal{P}_{\Pi^\perp}H_2\|_*,$$

$$|\langle H_2, F\rangle| = |\langle H_2, \mathcal{P}_{\Omega^\perp}F\rangle| = |\langle\mathcal{P}_{\Omega^\perp}H_2, F\rangle| \leq \|\mathcal{P}_{\Omega^\perp}H_2\|_1\|F\|_\infty \leq \frac{9}{10}\|\mathcal{P}_{\Omega^\perp}H_2\|_1,$$

$$|\langle H_2, \mathcal{P}_\Omega D\rangle| \leq |\langle\mathcal{P}_\Omega H_2, \mathcal{P}_\Omega D\rangle| \leq \frac{1}{4}\|\mathcal{P}_\Omega H_2\|_F.$$

Assume $\|R + E\|_F \leq \epsilon$, for all $i = 1, 2, \cdots, n$. Also note that

$$\|\mathcal{P}_\Omega H_2\|_F \leq \|\mathcal{P}_\Omega\mathcal{P}_\Pi H_2\|_F + \|\mathcal{P}_\Omega\mathcal{P}_{\Pi^\perp}H_2\|_F$$

$$\leq \frac{1}{4}\|H_2\|_F + \|\mathcal{P}_{\Pi^\perp}H_2\|_F$$

$$\leq \frac{1}{4}\|\mathcal{P}_\Omega H_2\|_F + \frac{1}{4}\|\mathcal{P}_{\Omega^\perp}H_2\|_F + \|\mathcal{P}_{\Pi^\perp}H_2\|_F,$$

then we have

$$\|\mathcal{P}_\Omega H_2\|_F \leq \frac{1}{3}\|\mathcal{P}_{\Omega^\perp}H_2\|_F + \frac{4}{3}\|\mathcal{P}_{\Pi^\perp}H_2\|_F \leq \frac{1}{3}\|\mathcal{P}_{\Omega^\perp}H_2\|_1 + \frac{4}{3}\|\mathcal{P}_{\Pi^\perp}H_2\|_*.$$

Plug into the previous inequality, also note that for $n_{(1)} \geq 16, \beta = \frac{1}{\sqrt{n_{(1)}}} \leq \frac{1}{4}$, it gives

$$2\lambda\epsilon\|H\|_F \geq \lambda\|H\|_F^2 - \sqrt{n_{(2)}}\|H\|_F - (\frac{9}{10} + \frac{\beta}{3})\|\mathcal{P}_{\Pi^\perp}H_2\|_* + \frac{\beta}{60}\|\mathcal{P}_{\Omega^\perp}H_2\|_1 + \|\mathcal{P}_{\Pi^\perp}H_1\|_*$$

$$\geq \lambda\|H\|_F^2 - \sqrt{n_{(2)}}\|H\|_F + \frac{\beta}{60}\|\mathcal{P}_{\Omega^\perp}H_2\|_1 + \frac{1}{60}\|\mathcal{P}_{\Pi^\perp}H_1\|_* + \frac{59}{60}(\|\mathcal{P}_{\Pi^\perp}H_1\|_* - \|\mathcal{P}_{\Pi^\perp}H_2\|_*)$$

$$= \lambda\|H\|_F^2 - \sqrt{n_{(2)}}\|H\|_F + \frac{\beta}{60}\|\mathcal{P}_{\Omega^\perp}H_2\|_1 + \frac{1}{60}\|\mathcal{P}_{\Pi^\perp}H_1\|_* + \frac{59}{60}(\|\mathcal{P}_{\Pi^\perp}H_1\|_* - \|\mathcal{P}_{\Pi^\perp}(-H_2)\|_*)$$

$$\geq \lambda\|H\|_F^2 - \sqrt{n_{(2)}}\|H\|_F + \frac{\beta}{60}\|\mathcal{P}_{\Omega^\perp}H_2\|_1 + \frac{1}{60}\|\mathcal{P}_{\Pi^\perp}H_1\|_* - \frac{59}{60}\|\mathcal{P}_{\Pi^\perp}(H_1 + H_2)\|_*$$

$$\geq \lambda\|H\|_F^2 - \sqrt{n_{(2)}}\|H\|_F + \frac{\beta}{60}\|\mathcal{P}_{\Omega^\perp}H_2\|_1 + \frac{1}{60}\|\mathcal{P}_{\Pi^\perp}H_1\|_* - \|H\|_*.$$

The last inequality is due to

$$\|\mathcal{P}_{\Pi^\perp}H\|_* = \sup_{\|X\|_2\leq 1}\langle\mathcal{P}_{\Pi^\perp}H, X\rangle = \sup_{\|X\|_2\leq 1}\langle H, \mathcal{P}_{\Pi^\perp}X\rangle \leq \sup_{\|\mathcal{P}_{\Pi^\perp}X\|_2\leq 1}\langle H, \mathcal{P}_{\Pi^\perp}X\rangle \leq \sup_{\|X\|_2\leq 1}\langle H, X\rangle = \|H\|_*.$$

Note that $\|H\|_* \leq \sqrt{n_{(2)}}\|H\|_F$, then we obtain

$$2\lambda\epsilon\|H\|_F \geq \lambda\|H\|_F^2 - 2\sqrt{n_{(2)}}\|H\|_F + \frac{\beta}{60}\|\mathcal{P}_{\Omega^\perp}H_2\|_1 + \frac{1}{60}\|\mathcal{P}_{\Pi^\perp}H_1\|_*.$$

Rewrite this inequality gives

$$\frac{\beta}{60}\|\mathcal{P}_{\Omega^\perp}H_2\|_1 + \frac{1}{60}\|\mathcal{P}_{\Pi^\perp}H_1\|_* \leq -\lambda\|H\|_F^2 + 2(\sqrt{n_{(2)}} + \lambda\epsilon)\|H\|_F$$

$$= -\lambda(\|H\|_F - (\frac{\sqrt{n_{(2)}} + \lambda\epsilon}{\lambda}))^2 + (\frac{\sqrt{n_{(2)}}}{\sqrt{\lambda}} + \sqrt{\lambda}\epsilon)^2$$

$$\leq (\frac{\sqrt{n_{(2)}}}{\sqrt{\lambda}} + \sqrt{\lambda}\epsilon)^2.$$

Recall that in our original optimization problem, we should consider above inequalities for the summation of all the local patches, denote $h_i \equiv \|H^{(i)}\|_F$, then

$$\sum_{i=1}^{n} \frac{\beta_i}{60} \|\mathcal{P}_{\Omega_i^{\perp}} H_2^{(i)}\|_1 + \frac{1}{60} \sum_{i=1}^{n} \|\mathcal{P}_{\Pi_i^{\perp}} H_1^{(i)}\|_*$$

$$\leq \sum_{i=1}^{n} -\lambda_i \|H^{(i)}\|_F^2 + 2\sqrt{\min\{k_i+1,p\}} \|H^{(i)}\|_F + 2\lambda_i \epsilon_i \|H^{(i)}\|_F$$

$$= \sum_{i=1}^{n} -\lambda_i h_i^2 + 2\sqrt{\min\{k_i+1,p\}} h_i + 2\lambda_i \epsilon_i h_i$$

$$= \sum_{i=1}^{n} -\lambda_i (h_i - \frac{\sqrt{\min\{k_i+1,p\}} + \lambda_i \epsilon_i}{\lambda_i})^2 + (\frac{\sqrt{\min\{k_i+1,p\}}}{\sqrt{\lambda_i}} + \sqrt{\lambda_i}\epsilon_i)^2$$

$$\leq 4 \sum_{i=1}^{n} \sqrt{\min\{k_i+1,p\}} \epsilon_i,$$

where we choose $\lambda_i = \frac{\sqrt{\min\{k_i+1,p\}}}{\epsilon_i}$, and $\beta_i = \frac{1}{\sqrt{\max\{k_i+1,p\}}}$ .

Then we have the bound for $\sum_{i=1}^{n} \|\mathcal{P}_{\Pi_i^{\perp}} H_1^{(i)}\|_*$ and $\sum_{i=1}^{n} \|\mathcal{P}_{\Omega_i^{\perp}} H_2^{(i)}\|_1$

$$\sum_{i=1}^{n} \|\mathcal{P}_{\Pi_i^{\perp}} H_1^{(i)}\|_* \leq C\sqrt{\min\{\bar{k},p\}} \sum_{i=1}^{n} \epsilon_i \leq C\sqrt{\min\{\bar{k},p\}}\sqrt{n}\|\epsilon\|_2,$$

$$\sum_{i=1}^{n} \|\mathcal{P}_{\Omega_i^{\perp}} H_2^{(i)}\|_1 \leq C\sqrt{\max_i \max\{k_i,p\}} \sum_{i=1}^{n} \sqrt{\min\{k_i,p\}} \epsilon_i$$

$$= C\sqrt{\max\{\bar{k},p\}} \sum_{i=1}^{n} \sqrt{\min\{k_i,p\}} \epsilon_i$$

$$\leq C\sqrt{\max\{\bar{k},p\}\min\{\bar{k},p\}} \sum_{i=1}^{n} \epsilon_i$$

$$\leq C\sqrt{p\bar{k}}\sqrt{n}\|\epsilon\|_2.$$

Denote $H_2^{(i)} \equiv \mathcal{P}_i(\hat{S}) - S^{(i)}$, to estimate the error bound of $\sum_{i=1}^{n} \|H_2^{(i)}\|_{2,1}$, we decompose $H_2^{(i)}$ into three parts, for each $i = 1, 2, \cdots n$

$$\|H_2^{(i)}\|_F \leq \|(I - \mathcal{P}_{\Omega_i})H_2^{(i)}\|_F + \|(\mathcal{P}_{\Omega_i} - \mathcal{P}_{\Omega_i}\mathcal{P}_{\Pi_i})H_2^{(i)}\|_F + \|\mathcal{P}_{\Omega_i}\mathcal{P}_{\Pi_i}H_2^{(i)}\|_F$$

$$\leq \|\mathcal{P}_{\Omega_i^{\perp}} H_2^{(i)}\|_F + \|\mathcal{P}_{\Pi_i^{\perp}} H_2^{(i)}\|_F + \frac{1}{4}\|H_2^{(i)}\|_F,$$

which leads to

$$\|H_2^{(i)}\|_F \leq \frac{4}{3}(\|\mathcal{P}_{\Omega_i^{\perp}} H_2^{(i)}\|_F + \|\mathcal{P}_{\Pi_i^{\perp}} H_2^{(i)}\|_F)$$

$$= \frac{4}{3}(\|\mathcal{P}_{\Omega_i^{\perp}} H_2^{(i)}\|_1 + \|\mathcal{P}_{\Pi_i^{\perp}} H_1^{(i)}\|_* + \|\mathcal{P}_{\Pi_i^{\perp}} H^{(i)}\|_F)$$

$$\leq \frac{4}{3}(\|\mathcal{P}_{\Omega_i^{\perp}} H_2^{(i)}\|_1 + \|\mathcal{P}_{\Pi_i^{\perp}} H_1^{(i)}\|_* + \|H^{(i)}\|_F).$$

Next, we need to bound $\sum_{i=1}^{n} \|H^{(i)}\|_F$, note that $\lambda_i = \frac{\sqrt{\min\{k_i+1,p\}}}{\epsilon_i}$ and

$$\sum_{i=1}^{n} -\lambda_i h_i^2 + 2\sqrt{\min\{k_i+1,p\}} h_i + 2\lambda_i \epsilon_i h_i \geq 0,$$

which gives

$$4\sqrt{\min\{\bar{k}+1,p\}}\sum_{i=1}^{n}h_i \geq 4\sum_{i=1}^{n}\sqrt{\min\{k_i+1,p\}}h_i \geq \sum_{i=1}^{n}\sqrt{\min\{k_i+1,p\}}\frac{h_i^2}{\epsilon_i} \geq \sqrt{\min\{\underline{k}+1,p\}}\sum_{i=1}^{n}\frac{h_i^2}{\epsilon_i},$$

by Cauchy inequality

$$\sum_{i=1}^{n}\frac{h_i^2}{\epsilon_i} \geq \frac{(\sum_{i=1}^{n}h_i)^2}{\sum_{i=1}^{n}\epsilon_i} \geq \frac{(\sum_{i=1}^{n}h_i)^2}{\sqrt{n}\|\epsilon\|_2},$$

then we obtain

$$\sum_{i=1}^{n}h_i \leq 4\sqrt{\frac{\min\{\bar{k}+1,p\}}{\min\{\underline{k}+1,p\}}}\sqrt{n}\|\epsilon\|_2 \leq C\sqrt{n}\|\epsilon\|_2,$$

the last inequality is due to $\frac{nq}{2} \leq \underline{k} \leq \bar{k} \leq 2nq$, which is guaranteed with high probability by Lemma 5, thus

$$\sum_{i=1}^{n}\|H_2^{(i)}\|_F \leq \frac{4}{3}(\sum_{i=1}^{n}\|\mathcal{P}_{\Omega_i^{\perp}}H_2^{(i)}\|_1 + \sum_{i=1}^{n}\|\mathcal{P}_{\Pi_i^{\perp}}H_1^{(i)}\|_* + \sum_{i=1}^{n}\|H^{(i)}\|_F)$$

$$= \frac{4}{3}(\sum_{i=1}^{n}\|\mathcal{P}_{\Omega_i^{\perp}}H_2^{(i)}\|_1 + \sum_{i=1}^{n}\|\mathcal{P}_{\Pi_i^{\perp}}H_1^{(i)}\|_* + \sum_{i=1}^{n}h_i)$$

$$\leq C\sqrt{p\bar{k}n}\|\epsilon\|_2.$$

Now let's divide $H_2^{(i)}$ into columns to get the $\ell_{2,1}$ norm error bound, denote $(H_2^{(i)})_j$ as the $j$th column in $H_2^{(i)}$, then we can derive the $\ell_{2,1}$ norm error bound in Lemma 4

$$C\sqrt{p\bar{k}n}\|\epsilon\|_2 \geq \sum_{i=1}^{n}\|H_2^{(i)}\|_F = \sum_{i=1}^{n}\sqrt{\sum_{j=1}^{k_i+1}\|(H_2^{(i)})_j\|_2^2}$$

$$\gtrsim \sum_{i=1}^{n}\sqrt{\frac{1}{k_i}(\sum_{j=1}^{k_i}\|(H_2^{(i)})_j\|_2)^2}$$

$$\gtrsim \frac{1}{\sqrt{\bar{k}}}\sum_{i=1}^{n}\sum_{j=1}^{k_i}\|(H_2^{(i)})_j\|_2.$$

Then we obtain

$$\sum_{i}\|\mathcal{P}_i(\hat{S}) - S^{(i)}\|_{2,1} = \sum_{i=1}^{n}\sum_{j=1}^{k_i+1}\|(H_2^{(i)})_j\|_2 \leq C\sqrt{pn\bar{k}}\|\epsilon\|_2.$$

□

**Lemma 5.** *If $qn \geq 9\log n$, with probability at least $1 - 2\exp(-c_3 qn)$, $\frac{qn}{2} \leq k_i \leq 2qn$, for all $i = 1, 2, \cdots, n$, here $c_3$ is some constants not related to $q$ and $n$.*

*Proof.* Since each $X_i$ is drawn from a manifold $\mathcal{M} \subseteq \mathbb{R}^p$ independently and uniformly, for some fixed $(\eta_{i_0}, q)$-neighborhood of $X_{i_0}$, for each $j = \{1, 2, \cdots, n\}\backslash\{i_0\}$, the probability that $X_j$ falls into $(\eta_{i_0}, q)$-neighborhood is $q$. Since $\{X_i\}_{i=1,2,\cdots n}$, $k_i$ follows i.i.d binomial distribution $B(n, q)$, we can apply large deviations inequalities to derive an upper and lower bound for $k_i$. By Theorem 1 in [2], we have that for each $i = 1, 2, \cdots, n$

$$\mathbb{P}(k_i > 2qn) \leq \exp(-\frac{(qn)^2}{2(qn(1-q) + qn/3)}) \leq \exp(-\frac{3}{8}qn),$$

$$\mathbb{P}(k_i < \frac{qn}{2}) \leq \exp(-\frac{(qn/2)^2}{2qn}) = \exp(-\frac{1}{8}qn).$$

Therefore by Union Bound Theorem

$$\mathbb{P}(\frac{qn}{2} \le k_i \le 2qn, \forall i = 1, 2, \cdots n) \ge 1 - n(\exp(-\frac{3}{8}qn) + \exp(-\frac{1}{8}qn))$$

$$\ge 1 - 2n\exp(-\frac{1}{8}qn)$$

$$= 1 - 2\exp(-\frac{1}{8}qn + \log n)$$

$$\ge 1 - 2\exp(-\frac{1}{72}qn).$$

□

## 1.2 Removing (2) and (3) in Assumption A

We will show that under our assumption that points are uniformly drawn from the manifold, (2) and (3) in Assumption A automatically hold provided (1) holds, thus they can be removed from the requirements.

Let us again restrict our attention to an individual patch and for the simplicity of notation, ignore the superscript $i$ (the treatment for all patches are the same). Recall that $\mathcal{C}(L) = \mathcal{C}(T) = U\Sigma V^*$, and $\tilde{V}$ is the orthonormal basis of $span([\mathbf{1}, V])$, since $0 = \mathcal{C}(T)\mathbf{1} = U\Sigma V^*\mathbf{1}$, we have $V^*\mathbf{1} = 0$, then $\mathbf{1} \perp span(V)$, thus we can write one basis for $span([\mathbf{1}, V])$ as $[\frac{1}{\sqrt{k+1}}\mathbf{1}, V]$, which indicates that in order to remove (2), we only need to show that with high probability, $V$ has small coherence. Also, recall that $T^{(i)} = P_{T_{X_i}}(X^{(i)} - X_i\mathbf{1}^T)$, since each $X_i$ is independent, each column in $T^{(i)}$ is also independent. In addition, each column is in the span of the tangent space with $U$ being an orthonormal basis. Therefore $T = U\Lambda \equiv U[\alpha_1, \alpha_2, ..., \alpha_k, 0]$, where $\alpha_i, i = 1, 2, \cdots, k$ is the $i$th column of $\Lambda$, which corresponds to the coefficients of the $i$th column of $T$ under $U$, the last column is zero vector since it corresponds to $X_i$ itself. Since columns of $T$ are i.i.d, then $\alpha_i$s are also i.i.d., so they all obey the same distribution as a random vector $\alpha$. We establish the following lemma for the right singular vectors of $T$.

**Lemma 6.** *Let $\mathcal{C}(T) = U\Sigma V^*$ be the reduced singular vector decomposition of $\mathcal{C}(T)$, assume $C \equiv \mathbb{E}((\alpha - \mathbb{E}\alpha)(\alpha - \mathbb{E}\alpha)^*)$ has a finite condition number. Then, with probability at least $1 - 2d\exp(-ck)$, the right singular vector $V$ obeys*

$$\max_{1 \le j \le k} \|V^*\mathbf{e}_j\|^2 \le \frac{c}{k},$$

*and with (1) in Assumption A*

$$\|UV^*\|_\infty \le \sqrt{\frac{cd}{pk}}.$$

*Proof.* As discussed above, $\mathcal{C}(T)$ has the following representation

$$\mathcal{C}(T) = TG = U[\alpha_1, \alpha_2, \cdots, \alpha_k, \mathbf{0}]G,$$

where $U \in \mathbb{R}^{p,d}$ is an orthonormal basis of the tangent space, and $\Lambda = [\alpha_1, \alpha_2, ..., \alpha_k, \mathbf{0}] \in \mathbb{R}^{d,k+1}$ is the coefficients of randomly drawn points in a neighbourhood projected to the tangent space.

Since points are randomly drawn from an neighbourhood contained in a ball of radius at most $\eta$, one can easily verify that $\|\alpha_j\|_2 \le \eta$ for each $j = 1, ..., k$. Assume $TG$ and $\Lambda$ have the reduced SVD of the form

$$TG = U\Sigma V^*, \ \Lambda G = U_\Lambda \Sigma_\Lambda V_\Lambda^*,$$

Then $T$ can be written as

$$TG = U\Sigma V^* = UU_\Lambda \Sigma_\Lambda V_\Lambda^*.$$

It can be verified that $\text{null}(TG)$ is the span of columns in $(V_\Lambda)^C$, then we have $span(V_\Lambda) = span(V)$, since both $V_\Lambda$ and $V$ are orthonormal, they are equal up to a rotation, i.e. $\exists R \in \mathbb{R}^{d,d}$, $R^*R = RR^* = I$, such that $V = V_\Lambda R$. Then

$$\max_{1 \le j \le k} \|V^*\mathbf{e}_j\|^2 = \max_{1 \le j \le k} \|R^*V_\Lambda^*\mathbf{e}_j\|^2 = \max_{1 \le j \le k} \|V_\Lambda^*\mathbf{e}_j\|^2.$$

Next we bound the coherence of $V_\Lambda$. Since $V_\Lambda^* = \Sigma_\Lambda^{-1} U_\Lambda^* \Lambda G$, we have

$$\max_{1 \leq j \leq k} \|\Sigma_\Lambda^{-1} U_\Lambda^* \Lambda G \mathbf{e}_j\| \leq \|\Sigma_\Lambda^{-1}\| \max_{1 \leq i \leq k} \|U_\Lambda^* \Lambda G \mathbf{e}_j\|$$

$$= \|\Sigma_\Lambda^{-1}\| \max_{1 \leq j \leq k} \|\Lambda G \mathbf{e}_j\|$$

$$\leq \|\Sigma_\Lambda^{-1}\| \max_{1 \leq j \leq k} \|\alpha_j - \bar{\alpha}\|$$

$$\leq 2\eta \|\Sigma_\Lambda^{-1}\|.$$

Recall that

$$\Lambda G = [\alpha_1, \alpha_2, \cdots, \alpha_k, 0](I - \frac{1}{k+1} \mathbf{1}\mathbf{1}^T)$$

$$= [\alpha_1 - \bar{\alpha}, \alpha_2 - \bar{\alpha}, \cdots, \alpha_k - \bar{\alpha}, -\bar{\alpha}]$$

$$= [\alpha_1 - \mathbb{E}\alpha, \alpha_2 - \mathbb{E}\alpha, \cdots, \alpha_k - \mathbb{E}\alpha, 0] - [\bar{\alpha} - \mathbb{E}\alpha, \bar{\alpha} - \mathbb{E}\alpha, \cdots, \bar{\alpha} - \mathbb{E}\alpha, \bar{\alpha}],$$

where $\bar{\alpha} = \frac{1}{k+1} \sum_{i=1}^k \alpha_i$, thus

$$|\sigma_d(\Lambda G) - \sigma_d([\alpha_1 - \mathbb{E}\alpha, \alpha_2 - \mathbb{E}\alpha, \cdots, \alpha_k - \mathbb{E}\alpha, 0])|$$

$$\leq \|[\bar{\alpha} - \mathbb{E}\alpha, \bar{\alpha} - \mathbb{E}\alpha, \cdots, \bar{\alpha} - \mathbb{E}\alpha, \bar{\alpha}]\|_2$$

$$\leq \|[\bar{\alpha} - \mathbb{E}\alpha, \bar{\alpha} - \mathbb{E}\alpha, \cdots, \bar{\alpha} - \mathbb{E}\alpha, \bar{\alpha} - \mathbb{E}\alpha]\|_2 + \|\mathbb{E}\alpha\|_2$$

$$\leq \sqrt{k+1} \|\frac{1}{k+1} \sum_{i=1}^k (\alpha_i - \mathbb{E}\alpha) - \frac{1}{k+1} \mathbb{E}\alpha\|_2 + \eta \tag{7}$$

$$\leq \|\frac{1}{\sqrt{k+1}} \sum_{i=1}^k (\alpha_i - \mathbb{E}\alpha)\|_2 + 2\eta$$

Fitst, we want to use Bernstein Matrix Inequality to bound the $\ell_2$-norm in the last inequality. Denote $\beta_i = \frac{1}{\sqrt{k+1}}(\alpha_i - \mathbb{E}\alpha)$, $Z = \sum_{i=1}^k \beta_i$, then $\beta_i$ is independent, we also have

$$\mathbb{E}\beta_i = 0, \ \|\beta_i\|_2 \leq \frac{1}{\sqrt{k+1}}(\|\alpha_i\|_2 + \|\mathbb{E}\alpha\|_2) \leq \frac{2\eta}{\sqrt{k}},$$

which means $\beta_i$ has mean zero and is uniformly bounded, also

$$\nu(Z) = \max\{\|\mathbb{E}(ZZ^*)\|_2, \|\mathbb{E}(Z^*Z)\|_2\}$$

$$= \max\{\|\sum_{i=1}^n \mathbb{E}(\beta_i \beta_i^*)\|_2, \|\sum_{i=1}^n \mathbb{E}(\beta_i^* \beta_i)\|_2\}$$

$$= \frac{k}{k+1} \max\{\|\mathbb{E}(\alpha_i - \mathbb{E}\alpha)(\alpha_i - \mathbb{E}\alpha)^T\|_2, \mathbb{E}\,\mathrm{tr}((\alpha_i - \mathbb{E}\alpha)(\alpha_i - \mathbb{E}\alpha)^T))\}$$

$$< \max\{\|C\|_2, \mathrm{tr}(C)\}$$

$$< d\sigma_1(C).$$

By assumption, $C$ has finite condition number, and $d \ll k$, by Matrix Bernstein inequality, we are able to bound the spectral norm of $Z$

$$P(\|Z\|_2 \geq t) \leq (d+1)\exp(\frac{-t^2}{d\sigma_1(C) + \frac{2\eta t}{3\sqrt{k}}})$$

Let $t = \frac{\sqrt{\sigma_d(C)k}}{4}$, we have

$$P(\|Z\|_2 \geq \frac{\sqrt{\sigma_d(C)k}}{4}) \leq d\exp(-ck). \tag{8}$$

Next, equipped with Matrix Bernstein inequality again, we can prove that $\sigma_d([\alpha_1 - \mathbb{E}\alpha, \alpha_2 - \mathbb{E}\alpha, \cdots, \alpha_k - \mathbb{E}\alpha, 0])$ concentrates around $\sigma_d(C)$. Note that $\sigma_d^2([\alpha_1 - \mathbb{E}\alpha, \alpha_2 - \mathbb{E}\alpha, \cdots, \alpha_k - \mathbb{E}\alpha, 0]) = \sigma_d(\sum_{i=1}^k (\alpha_i - \mathbb{E}\alpha)(\alpha_i - \mathbb{E}\alpha)^T)$, we consider

$$|\sigma_d(\sum_{i=1}^k (\alpha_i - \mathbb{E}\alpha)(\alpha_i - \mathbb{E}\alpha)^T) - k\sigma_d(C)| \leq \|\sum_{i=1}^n (\alpha_i - \mathbb{E}\alpha)(\alpha_i - \mathbb{E}\alpha)^T - kC\|_2$$

Similar as what we discussed above, let $Z_j = (\alpha_j - \mathbb{E}\alpha)(\alpha_j - \mathbb{E}\alpha)^T - C$, $j = 1, 2, \cdots, k$. It can be verified that $Z_j$ is bounded

$$\|Z_j\|_2 \leq \|\alpha_j - \mathbb{E}\alpha\|_2^2 + \sigma_1(C) \leq 2\eta^2 + \sigma_1(C) \equiv c_4.$$

Since $Z_j$ follows i.i.d distribution, we also have $\nu(Z) \leq kc_5$ for some constant $c_5$ which represents the variance of $Z_j$. Applying matrix Bernstein inequality, we obtain

$$\mathbb{P}\Big(\|\sum_{j=1}^{k}(\alpha_j - \mathbb{E}\alpha)(\alpha_j - \mathbb{E}\alpha)^T - kC\|_2 \geq t\Big) \leq 2d\exp(-\frac{t^2}{kc_5 + \frac{c_4 t}{3}})$$

further, take $t = \frac{3k\sigma_d(C)}{4}$, then with probability over $1 - 2d\exp(-c_6 k)$ for some constant $c_6$, the following holds

$$|\sigma_d(\sum_{i=1}^{k}(\alpha_i - \mathbb{E}\alpha)(\alpha_i - \mathbb{E}\alpha)^T) - k\sigma_d(C)| \leq \|\sum_{i=1}^{n}(\alpha_i - \mathbb{E}\alpha)(\alpha_i - \mathbb{E}\alpha)^T - kC\|_2 < \frac{3k\sigma_d(C)}{4},$$

which leads to

$$\sigma_d^2([\alpha_1 - \mathbb{E}\alpha, \alpha_2 - \mathbb{E}\alpha, \cdots, \alpha_k - \mathbb{E}\alpha]) = \sigma_d(\sum_{i=1}^{k}(\alpha_i - \mathbb{E}\alpha)(\alpha_i - \mathbb{E}\alpha)^T) > \frac{k\sigma_d(C)}{4},$$

thus

$$\sigma_d([\alpha_1 - \mathbb{E}\alpha, \alpha_2 - \mathbb{E}\alpha, \cdots, \alpha_k - \mathbb{E}\alpha]) > \frac{\sqrt{k\sigma_d(C)}}{2}. \tag{9}$$

Combine (7), (8) and (9), we have proved that with probability at least $1 - d\exp(-ck)$, $\sigma_d(\Lambda P) \succsim \sqrt{k}$, therefore $\|\Sigma_\Lambda^{-1}\| \precsim \frac{1}{\sqrt{k}}$, which further gives $\max_{1 \leq j \leq k+1} \|V^* \mathbf{e}_j\|^2 \precsim \frac{1}{k}$.

Finally, with (1) in Assumption A, (3) is also satisfied with the same probability, since

$$\|UV^*\|_\infty \leq \max_j \|U^* \mathbf{e}_j\|_2 \max_l \|V^* \mathbf{e}_l\|_2 \leq \sqrt{\frac{cd}{pk}}.$$

Hence (3) in Assumption A can also be removed. $\square$

The above discussion is valid for each patch individually, i.e., with probability at least $1 - d\exp(-ck_i) \geq 1 - d\exp(-c\underline{k})$, (2) and (3) hold for any fixed $i = 1, 2, \cdots n$. By union bound inequality, with probability at least $1 - nd\exp(-c\underline{k})$, (2) and (3) hold for all the local patches.

Note that $1 - nd\exp(-c\underline{k}) = 1 - \exp(-c\underline{k} + \log n)$, here we omit $d$ since it is very small. By Lemma 5, with probability at least $1 - 2\exp(-c_1 qn)$, $\frac{nq}{2} \leq k_i \leq 2nq$, for all $i = 1, 2, \cdots n$. Using the assumption in Theorem 4.2, $qn \geq c_2 \log n$ for some constant $c_2$ larger enough, we can see that with probability over $1 - \exp(-c_3 k)$, the requirement (2) and (3) automatically hold due to i.i.d assumption on the samples, which enable us to remove these assumptions in Theorem 4.2.

## 1.3 Proof of the convergence of $\frac{\hat{\lambda}_i - \lambda_i^*}{\lambda_i^*}$ as $k \to \infty$

When $k$ is large enough, $\min\{k+1, p\} = p$, $\hat{\lambda}_i = \frac{\sqrt{p}}{\hat{\epsilon}_i}$, $\lambda_i^* = \frac{\sqrt{p}}{\epsilon_i}$, then

$$\frac{\hat{\lambda}_i - \lambda_i^*}{\lambda_i^*} = \frac{\frac{\sqrt{p}}{\hat{\epsilon}_i} - \frac{\sqrt{p}}{\epsilon_i}}{\frac{\sqrt{p}}{\epsilon_i}} = \frac{\epsilon_i - \hat{\epsilon}_i}{\hat{\epsilon}_i} = \frac{\epsilon_i}{\hat{\epsilon}_i} - 1.$$

In order to show $|\frac{\hat{\lambda}_i - \lambda_i^*}{\lambda_i^*}| \xrightarrow{k\to\infty} 0$, it is sufficient to prove that $\frac{\epsilon_i^2 - \hat{\epsilon}_i^2}{\hat{\epsilon}_i^2} = \frac{\epsilon_i^2}{\hat{\epsilon}_i^2} - 1 \xrightarrow{k\to\infty} 0$, thus $\frac{\epsilon_i}{\hat{\epsilon}_i} \xrightarrow{k\to\infty} 1$, hence $\frac{\hat{\lambda}_i - \lambda_i^*}{\lambda_i^*} \xrightarrow{k\to\infty} 0$. Notice that

$$\left| \frac{\epsilon_i^2 - \hat{\epsilon}_i^2}{\hat{\epsilon}_i^2} \right| = \left| \frac{\|R^{(i)} + N^{(i)}\|_F^2 - \left((k+1)p\sigma^2 + \sum_{j=1}^{k} \frac{\|X_i - X_{i_j}\|_2^4}{4} \bar{\Gamma}^2(X_i)\right)}{(k+1)p\sigma^2 + \sum_{j=1}^{k} \frac{\|X_i - X_{i_j}\|_2^4}{4} \bar{\Gamma}^2(X_i)} \right|$$

$$\leq \left| \frac{\left(\|N^{(i)}\|_F^2 - (k+1)p\sigma^2\right) + \left(\|R^{(i)}\|_F^2 - \sum_{j=1}^{k} \frac{\|X_i - X_{i_j}\|_2^4}{4} \bar{\Gamma}^2(X_i)\right) + \langle N^{(i)}, R^{(i)} \rangle}{kp\sigma^2} \right|$$

$$\leq \left| \frac{\|N^{(i)}\|_F^2 - (k+1)p\sigma^2}{kp\sigma^2} \right| + \left| \frac{\|R^{(i)}\|_F^2 - \sum_{j=1}^{k} \frac{\|X_i - X_{i_j}\|_2^4}{4} \bar{\Gamma}^2(X_i)}{kp\sigma^2} \right| + \left| \frac{\sum_{j=1}^{k} \langle N_j^{(i)}, R_j^{(i)} \rangle}{(k+1)p\sigma^2} \right|.$$

Since each entry in $N^{(i)}$ follows i.i.d. obeying $\mathcal{N}(0, \sigma^2)$, $\langle N_j^{(i)}, R_j^{(i)} \rangle$ are also i.i.d. with $\mathbb{E}(\langle N_j^{(i)}, R_j^{(i)} \rangle) = 0$, by law of large numbers, the first and third term approximates 0 when $k \to \infty$. Also, by (12) and (13) in §5, the second term also approximates 0, thus $\frac{\epsilon_i^2 - \hat{\epsilon}_i^2}{\hat{\epsilon}_i^2} \xrightarrow{k\to\infty} 0$.

## 2 More numerical simulations

### 2.1 High dimensional Swiss roll

In the main paper, we demonstrated the superior performance of NRPCA on the 3D Swiss roll under the mixed noise model. We carried out the same simulation on a high dimension Swiss roll, and obtained better distinguishability among 1)-3). We also observed an overall improvement of the performance of NRPCA, which matches our intuition that the assumptions of Theorem 4.2 are more likely to be satisfied in high dimensions. The denoised results are displayed in Figure 1, where we clearly see that the neighbour update step effectively reduced more sparse noise, and the use of $\hat{X}$ instead of $\tilde{X} - \hat{S}$ allows a significant amount of Gaussian noise to be removed from the data.

In the high dimensional simulation, we generated a Swiss roll in $\mathbb{R}^{20}$ as following:

1. Choose the number of samples $n = 2000$;

2. let $t$ be the vector of length $n$ containing the $n$ uniform grid points in the interval $[0, 4\pi]$ with grid space $4\pi/(n-1)$;

3. Set the first three dimensions of the data the same way as the 3D Swiss roll, for $i = 1, ..., n$,

$$X_i(1) = (t(i) + 1)\cos(t(i));$$
$$X_i(2) = (t(i) + 1)\sin(t(i));$$
$$X_i(3) \sim \text{unif}([0, 8\pi]),$$

where $\text{unif}([0, 8\pi])$ means the uniform distribution on the interval $[0, 8\pi]$.

4. Set the 4-20 dimensions of the data to contain pure sinusoids with various frequencies

$$X_i(k) = t(i)\sin(f_k t(i)), \quad k = 4, ..., 20, .$$

where $f_k = k/21$ is the frequency for the $k$th dimension. The noisy data is obtained by adding i.i.d. Gaussian noise $\mathcal{N}(0, 0.25)$ to each entry of $X$ and adding sparse noise to 600 randomly chosen entries where the noise added to each chosen entry obeys $\mathcal{N}(5, 0.09)$.

### 2.2 MNIST

We observe some interesting dimension reduction results of the MNIST dataset with the help of NRPCA. It is well-known that the handwritten digits 4 and 9 have so high a similarity that some

Figure 1: NRPCA applied to the noisy 20D Swiss roll data set. $\tilde{X} - \hat{S}$ is the result after subtracting the estimated sparse noise via NRPCA with $T = 1$ "$\tilde{X} - \hat{S}$ with one neighbor update" is that with $T = 2$, i.e., patches are reassigned once; $\hat{X}$ is the denoised data obtained via fitting the tangent spaces in NRPCA with $T = 2$; "Patch-wise Robust PCA" refers to the ad-hoc application of the vanilla RPCA to each local patch independently, whose performance is clearly worse than the proposed joint-recovery formulation.

popular dimension reduction methods, such as Isomap and Laplacian Eigenmaps (LE) are not able to separate them into two clusters (first column of Figure 2). Despite the similarity, a few other methods (such as t-SNE) are able to distinguish them to a much higher degree, which suggests the possibility of improving the results of Isomap and LE with proper data pre-processing. We conjecture that the overlapping parts in Figure 2 (the left column) are caused by personalized writing styles with different beginning or finishing strokes. This type of differences can be better modelled by sparse noise than Gaussian or Poisson noises.

Figure 2: Laplacian eigenmaps and Isomap results for the original and the NRPCA denoised digits 4 and 9 from the MNIST dataset.

The right column of Figure 2 confirms this conjecture: after the NRPCA denoising (with $k = 6$), we see a much better separability of the two digits using the first two coordinates of Isomap and

Laplacian Eigenmaps. Here we used 2000 randomly drawn images of 4 and 9 from the MNIST training dataset. Figure 2 in the main paper used another random set of the same cardinally, but they both demonstrated that the denoising step greatly facilitates the dimensionality reduction.

In addition, we observe some emerging trajectory (or skeleton) patterns in the plot of the denoised embedding (right column of Figure 2). Mathematically speaking, this is due to the nuclear norm penalty on the tangent spaces in the optimization formulation that forces the denoised data to have a small intrinsic dimension. However, since the small intrinsic dimensionality is not manually inputted but implicitly imposed via an automatic calculation of the data curvature and the weight parameter $\lambda_i$, we do not think the trajectory pattern is a human artifact. To further examine the meaning the trajectories, we replaced the dots in the bottom two scattered plots in Figure 2 by their original images of the digits, and obtained Figure 3 and Figure 4. We can see that 1). the digits are better grouped in the denoised embedding than the orignal one and 2). the trajectories in the denoised embedding correspond to graduate transitions between the two images on the two ends. If two images are connected by two trajectories, then it indicates two ways for one image to gradually deform into the other. Furthermore, Figure 5 listed a few images of 4 and 9 before and after denoising, which shows which part of the image is detected as sparse noise and changed by NRPCA.

Figure 3: Isomap embedding using the original data from the MNIST dataset.

Figure 4: Isomap embedding using the Denoised data via NRPCA.

Original images for digit 4

Denoised images for digit 4

Original images for digit 9

Denoised images for digit 9

Figure 5: A comparison of the original and the NRPCA denoised images of digit 4 and 9.

Figure 6 shows the results for NRPCA denoising with more iterations of patch-reassignment, we can see that the results almost have no visible difference after $T > 2$. Since the patch-reassignment is in the outer iteration, increasing its frequency greatly increases the computation time. Fortunately, we find that often times two iterations are enough to deliver a good denoising result.

Noisy images

Denoised images with T=1

Denoised images with T=2

Denoised images with T=3

Denoised images with T=4

Denoised images with T=5

Figure 6: NRPCA Denoising results with more iterations of patch-reassignment.