[Reviews · NeurIPS 2019]

Reviewer 1



After rebuttal: I would like to thank the authors for addressing the comments. Their responses helped clarify some of my questions and helped better understand the paper, and therefore I am glad to increase my score. I am glad that the authors have decided to add the derivations in Sect. 6 because I think they are highly relevant. There are a few other things that I hope the authors will address for the final version of the paper: 1. Limitations of the method 2. Short introduction to RPCA 3. Related work as mentioned in the initial review. 4. A discussion on correcting vs. identifying outliers. 5. The discussion on kNN vs \epsNN on stability from the authors’ response is very helpful and would be useful to add it to the paper, otherwise it’s still not clear why wouldn’t the method use all the points in \epsNN (within radius r1). 6. Is the neighborhood size \eta the same for all points? What are its implications? 7. The part of the paper that still leaves me unsure are the experiments. Even if tSNE is not sensitive to outliers, I would still like to see results with tSNE in Fig 3. Still not convinced how to interpret the gaps that arise after denoising with NRPCA. Are they an artifact or reflect the real structure of the data? Even if they are an artifact, this does not discard the method, but a discussion would be useful (for example how do different values of the parameters affect the size of the gaps). 8. In the response the authors mention that they compare with LLE and Isomap, but they also used the Laplacian in Fig. 3 (I hope the difference between all these methods is clear to the authors, and is also addressed in one of my earlier comments as neither LLE not Isomap use the graph Laplacian). 9. If using more iterations than T=2 does not add any information, this should be discussed and explained. 10. There are also other small comments from the initial review that would need to be taken into account. Thank you. ============================================== Originality: The paper proposes a new method for nonlinear RPCA and the approach based on curvature estimation is interesting, but I feel the related literature is not always correctly and sufficiently cited. Quality: The paper is rigorous and provides both theoretical proofs and experimental results on both artificial and real-world data. A more thorough discussion of the weaknesses of the proposed method would be highly appreciated and useful. Clarity: The paper is clearly written and easy to read. Most of the notions introduced are well explained and the paper is well organized. Significance: Extending RPCA to nonlinear settings is an important research topic and could prove to have a high impact in practical applications. However I am not fully convinced by the experiments and some more detailed comments are presented below. General comments/questions: • A short introduction to RPCA would be useful for readers who are not very familiar with this technique and to see how NRPCA connects to RPCA. • There is some work on nonlinear settings connected to RPCA that the paper would need to make the connection to and possibly compare with: o B. Dai et al 2018-Connections with robust PCA and the role of emergent sparsity in variational autoencoder models o Y. Wang et al – Green generative models • Overall, I would have liked to see a more detailed description of the optimization problem from Sect. 6 and maybe a bit less on the curvature estimation especially sect. 5.1 which introduces a lot of notation that is not used afterwards (could maybe go into the supplementary material). The closed-form in eq (13) is not straightforward, especially introducing the soft-thresholding operator. • The paper seems to take the approach that outliers need to be corrected, but I believe this depends on the application, and a discussion on correcting vs. identifying outliers would be relevant. How can we distinguish between these two very different tasks? Detailed comments: • Line 19: “finding the nodes of the graph” – not sure I understand • Line 20: The references cited, as least 16 (LLE) and 17 (Isomap) (I am not familiar with 14) do not use the graph Laplacian. Isomap uses multidimensional scaling to do the embedding, not the Laplacian. • Line 21: the literature on outlier detection seems quite old. • Does the methodology presented in Sect 2 work for non-Gaussian noise too? • Lines 55-61: is the neighbourhood patch defined with respect to X_i or \tilde{X}_i? I would believe it should be the noisy data \tilde{X}_i, but maybe I am missing something? • Line 57: consisting • Sect 5.2: I am not sure I understand the need to use \epsNN instead of kNN if anyway afterwards we have to pick randomly m points within the \epsNN? This is related to a more general comment: \epsNN is known to be difficult in very high dimensions where because of the curse of dimensionality almost all points tend to fall within a small neighbourhood and points tend to be equidistant. Why do the authors choose randomly m points within \epsNN instead of either using directly kNN with k=m or use all the points within \epsNN? • Is the neighbourhood size \eta the same for all points? Could this be a problem if the manifold is not sampled uniformly and the density varies? • The notations for the radius and the residual term – would help if they were different instead of having both as R. Maybe small r for the radius? • Eq (11) is a bit confusing as it uses both the X_i, X_i_j and p,q notations in the same equation and even the same term. Does the right term have \Sigma^2 (similar in eq (12))? • Would be good to have a derivation of (12) in the supplementary material. • The approximation in (11) seems to work for k sufficiently large, but that would include all the points in the limit. Also, this brings me back to the discussion on \epsNN vs kNN: if we need a very large k, why first do an \epsNN and then pick randomly m points? • Sect 6: writing L^i as a function of S is not straightforward and a more detailed derivation would be useful. If I understand correctly from Alg 1 the procedure in Sect 6 is iterative. If so, this should be mentioned and also explain how you choose the number of steps T or if there is a stopping criterion. • Lines 208-210: How could we know when the neighborhood is wrong because of the strong sparse noise in order to apply the solution outlined? • Line 216: is p=3? • In Fig. 1 I would start with the noisy data which is the input as in Fig. 4. Adding it at the very end is a bit counterintuitive. If I understand well the authors apply Alg 1 either with one iteration T=1 or with two iterations T=2? What happens for larger T? Usually iterative algorithms run until some criterion is fulfilled, with T >> 2. • Line 224: no reference to Laplacian eigenmaps, and was not cited either in the introduction. • Fig. 3: t-SNE is known to work very well on MNIST. A comparison would be useful. How do you explain the gaps in the denoised versions? Maybe a classification task would help to reveal how much the proposed method works better compared to existing approaches. What is the dimension of the NRPCA space, is it two? Could you also show those results if dim(NRPCA) = 2 like in Fig. 1? Could you please provide the parameters used for Laplacian eigenmaps and Isomap for the original vs. denoised versions? • Fig. 4: LLE is applied in the space of NRPCA if I understand correctly. What is the dimension of the NRPCA space, 2D? • It seems that NRPCA can be used either independently (just like any dimension reduction method) or as a preprocessing to other methods (LLE, Isomap etc). Would be useful to state this somewhere in the paper. How many iterations are used here for each figure? If I understand correctly, T=1 for second plot, and T=2 for the two rightmost plots. I’m still wondering what happens for larger T as in a previous comment? • Line 258: equally

Reviewer 2



Originality: this work is a combination of know methods with a completely new analysis on how to tune the parameters. Furthermore, in order to use the theory to estimate the parameters, the authors proposed a method based on Riemannian geometry. However, some related work should have be cited like [1-5] where there is indication on the guarantees of recovery and the complexity of the problem. Quality: the RPCA part is sound. However, the Riemannian geometry part is confusing. Some part of the proposed method (section 5.2) seems difficult to implement without more information. For example, how to pick pairs (at a given distance) on the manifold ? How the geodesic distance is truly computed ? These part are hidden is the proposed algorithm page 6. Moreover, I would have expect some clues about the guarantees of recovery (not necessarily a theorem). Clarity: most of the paper is well written. Still some parts lack motivations. For example at section 2, the presentation of the problematic is too fast and one key idea (the construction of the patches) is a little hidden on the text and lacks motivations (and citations like [6]). The section 5, which is perhaps the most important, is difficult to follow and some crucial information are missing such as how to pick points on the manifolds (a little algorithm could useful). In section 6, again some details are missing, for example equation (15) is not trivial to compute written this way. Significance: principal component analysis is one of "classical" analysis tool with dealing with data. This method proposes a robust framework able to deal with spurious data and the manifold regularization allows a good reconstruction. This is very important when looking for trends with PCA. [1] Wright, J., Ganesh, A., Rao, S., Peng, Y., & Ma, Y. (2009). Robust principal component analysis: Exact recovery of corrupted low-rank matrices via convex optimization. In Advances in neural information processing systems (pp. 2080-2088). [2] Chandrasekaran, V., Sanghavi, S., Parrilo, P. A., & Willsky, A. S. (2011). Rank-sparsity incoherence for matrix decomposition. SIAM Journal on Optimization, 21(2), 572-596. [3] Tao, M., & Yuan, X. (2011). Recovering low-rank and sparse components of matrices from incomplete and noisy observations. SIAM Journal on Optimization, 21(1), 57-81. [4] Gillis, N., & Vavasis, S. A. (2018). On the complexity of robust pca and ℓ 1-norm low-rank matrix approximation. Mathematics of Operations Research, 43(4), 1072-1084. [5] Candès, E. J., Li, X., Ma, Y., & Wright, J. (2011). Robust principal component analysis?. Journal of the ACM (JACM), 58(3), 11. [6] Vaksman, G., Zibulevsky, M., & Elad, M. (2016). Patch ordering as a regularization for inverse problems in image processing. SIAM Journal on Imaging Sciences, 9(1), 287-319.

Reviewer 3



1. While the basic idea is not so exciting, the proposed technique is well justified and the paper is well written. I would suggest to accept the paper if the space is enough. 2. In the presence of gross errors, it is not easy to estimate the true neighborhood accurately. But the proposed method explicitly depends on the estimated neighborhoods. It would be better to analyze the influence of the kNN structure estimated from the given data. 3. The following closely related paper should be included into discussions: Matrix Recovery with Implicitly Low-Rank Data, 2019. 4. The coherence defined in 4.1 had been defined and investigated in many other papers, e.g., Exact Matrix Completion via Convex Optimization, 2009. Low-rank Matrix Completion in the Presence of High Coherence, 2016. 5. For empirical validations, it would be better to show some results that the proposed technique can improve the accuracy of classification or clustering.

Reviewer 4



From a quick reading, it seems like a solid paper and a reasonable contribution. A better justification/stronger experimental results for the sparse noise model would improve the paper. I did not verify the details of the theoretical analyses. Line 246: "Embeddin" --> Embedding Laplacian Eigenmaps is used and discussed but not cited. A citation needs to be added.

[Author Response · NeurIPS 2019]

We thank all the reviewers for their time and effort. We appreciate the constructive feedback as well as the acknowl-
edgment of the significance of this work in the review reports. Let us summarize the main contributions of this work
as 1) extending the well known Robust PCA denoising technique to the manifold setting thus greatly broadened the
applications and 2) providing a solid theoretical guarantee for the method. The key observation is that the success
of the proposed method depends solely on the intrinsic property of the data manifold instead of specific sampling
procedures (Theorem 4.2), which makes our extension non-trivial. Last but not least, to avoid the hassle of choosing
tuning parameters, we proposed a curvature estimation method that could be useful in other contexts.

We are particularly grateful for the suggestion of the reviewers about Section 5-6. We will restructure these two sections
for clarity. Specifically, we will move Sect. 5.1 (A short review of related concepts in Riemannian geometry) to the
appendix, and use the released space to better explain the curvature estimation idea (e.g., add derivation of Eq. (12),
add explanations of the parameters in Figure 1 and their relation to those defined in the context of Sect. 5.2 and Sect.
5.3, and summarize the curvature estimation procedures in a small algorithm). We will also follow Reviewer 1's and
Reviewer 2's advice to add the derivation of Eq. (13) and Eq. (15).

Due to space limitations, below we only address the major concerns raised by the reviewers.

**About the curvature estimation method**: we apologize for not including enough details in the description of the
proposed curvature estimation method. We agree with Reviewer 2 that considering its importance, we should make
room to better explain this idea. Generally speaking, there are indeed parameters to be set in the curvature estimation
step, but the main algorithm (Algorithm 1) is rather *insensitive* to the choice of these parameters. Specifically, in
Sect. 5.2, we explained how to estimate the average curvature at each data point (Eq. (8)) which is used later to set
the parameter $\lambda_i$ in the NRPCA formulation (for completeness, we also mentioned how the same idea can be used
to estimate the overall curvature of manifold in Line 171-174, but the overall curvature is *not* used in the proposed
method). When estimating the curvature at some point $p$, our method requires (Sect. 5.2) choosing $n$ other points
independently and uniformly at random from a neighborhood of $p$ (say the neighborhood has a radius $r_1$), compute the
curvatures of the geodesic curves joining $p$ and these $n$ neighboring points using Eq. (7), and then take the average of
the computed curvatures to derive Eq. (8). During this process, we need to pre-set the aforementioned parameters $r_1$
and $n$, as well as the size $r_2$ of the $k$NN in the Dijkstra's algorithm used to compute the geodesic distances (mentioned
in Line 165). However, through numerical experiments, we found that the final result of the main algorithm (Algorithm
1) was very robust to different choices of all these parameters. We will include in the paper this remark as well as some
numerical experiments to justify the claimed stability of Algorithm 1 w.r.t. the choices of parameters. We are also able
to theoretically justify this approach under the ideal uniform sampling assumption.
**About $k$NN and $\epsilon$-neighborhood**: we thank Reviewer 1 for raising this issue. $k$NN is used in the actual implementation
of the proposed algorithm while $\epsilon$-neighborhood is used in establishing Theorem 4.2 (Line 105). This is a common
practice in manifold learning (e.g., in the proof of the convergence of the graph Laplacian to the Laplace-Beltrami
operator of the manifold [1]), as the mathematical treatment for the $\epsilon$-neighborhood is much easier than $k$NN, while
the implementation of $k$NN is more stable than $\epsilon$-neighborhood. More importantly, the performance of these two are
similar to each other under the uniform and sufficient sampling assumption.
**Does the methodology presented in Section 2 work for non-Gaussian noise too?** Answer: The theoretical result
does not rely on the distribution of noise, so the proposed method also works when noise is non-Gaussian, as long as its
magnitude is still small.
**What is the dimension of the NRPCA space, is it two?** Answer: Similar to Robust PCA, the output (i.e., the
denoised data matrix) of NRPCA is of the same size as the input (i.e., the noisy data matrix). That is to say, NRPCA
does not reduce the dimension of the data and is a pure denoising technique.
**Classification results based on LLE and Isomap**: In the numerical experiments, we conducted LLE and Isomap on
the NRPCA denoised data to see how the denoising affects the performances of LLE and Isomap. We used LLE and
Isomap instead of tSNE because they are much more sensitive to outliers and Gaussian noise, thus are good indicators
of whether or not the noises are successfully removed. As recommended by reviewer 1, we implemented a classification
task on digits 4 and 9 from MNIST dataset, based on the denoised data in the numerical section. We first applied Isomap
to the denoised data, then used SVM with Gaussian kernel for classification. The cross-validated classification error rate
is 6.75%, while the error rate is 22.30% when applying SVM to the 2D data embedded by Isomap from the original
noisy data, which indicates that our method effectively removed the noise, and the dimension reduction under Isomap
became better.
**Is the neighborhood patch defined with respect to $X_i$ or $\tilde{X}_i$? What happens for $T \gg 2$?** Answer: the initial
neighborhood patch is defined w.r.t. the noisy data. For $T > 1$, patches are updated along with the variables, and as
$T \to \infty$ converges to (hopefully) the neighborhood patches corresponding to the clean data. In our experiment, we did
see a trend of convergence every time when $T$ gets larger. The reason we only choose $T = 1, T = 2$ in the figures is
that the results do not change much after $T > 2$.

[1] Hein et al., "*From graphs to manifolds–weak and strong pointwise consistency of graph Laplacians*," 2005.


[Meta-Review · NeurIPS 2019]

The reviewers are in agreement that this is an original and sufficient contribution that can be accepted.